# Generative Fractional Diffusion Models

**Gabriel Nobis** *
Fraunhofer HHI

**Maximilian Springenberg**
Fraunhofer HHI

**Marco Aversa**
Dotphoton

**Michael Detzel**
Fraunhofer HHI

**Rembert Daems**
Ghent University
FlandersMake–MIRO

**Roderick Murray-Smith**
University of Glasgow

**Shinichi Nakajima**
BIFOLD, TU Berlin
RIKEN AIP

**Sebastian Lapuschkin**
Fraunhofer HHI

**Stefano Ermon**
Stanford University

**Tolga Birdal**
Imperial College London

**Manfred Opper**
TU Berlin
University of Potsdam
University of Birmingham

**Christoph Knochenhauer**
Technical University of Munich

**Luis Oala**
Dotphoton

**Wojciech Samek**
Fraunhofer HHI
TU Berlin
BIFOLD

## Abstract

We introduce the first continuous-time score-based generative model that leverages fractional diffusion processes for its underlying dynamics. Although diffusion models have excelled at capturing data distributions, they still suffer from various limitations such as slow convergence, mode-collapse on imbalanced data, and lack of diversity. These issues are partially linked to the use of light-tailed Brownian motion (BM) with independent increments. In this paper, we replace BM with an approximation of its non-Markovian counterpart, fractional Brownian motion (fBM), characterized by correlated increments and Hurst index $H \in (0, 1)$, where $H = 0.5$ recovers the classical BM. To ensure tractable inference and learning, we employ a recently popularized Markov approximation of fBM (MA-fBM) and derive its reverse-time model, resulting in *generative fractional diffusion models* (GFDM). We characterize the forward dynamics using a continuous reparameterization trick and propose *augmented score matching* to efficiently learn the score function, which is partly known in closed form, at minimal added cost. The ability to drive our diffusion model via MA-fBM offers flexibility and control. $H \leq 0.5$ enters the regime of *rough paths* whereas $H > 0.5$ regularizes diffusion paths and invokes long-term memory. The Markov approximation allows added control by varying the number of Markov processes linearly combined to approximate fBM. Our evaluations on real image datasets demonstrate that GFDM achieves greater pixel-wise diversity and enhanced image quality, as indicated by a lower FID, offering a promising alternative to traditional diffusion models[1].

## 1 Introduction

Recent years have witnessed a remarkable leap in generative diffusion models [1, 2, 3], celebrated for their ability to accurately learn data distributions and generate high-fidelity samples. These models have made significant impact across a wide spectrum of application domains, including the generation of complex molecular structures [4] for material [5] or drug discovery [6], realistic

---

*Corresponding author gabriel.nobis@hhi.fraunhofer.de

[1]The implementation of our framework is available at `https://github.com/GabrielNobis/gfdm`.

38th Conference on Neural Information Processing Systems (NeurIPS 2024).

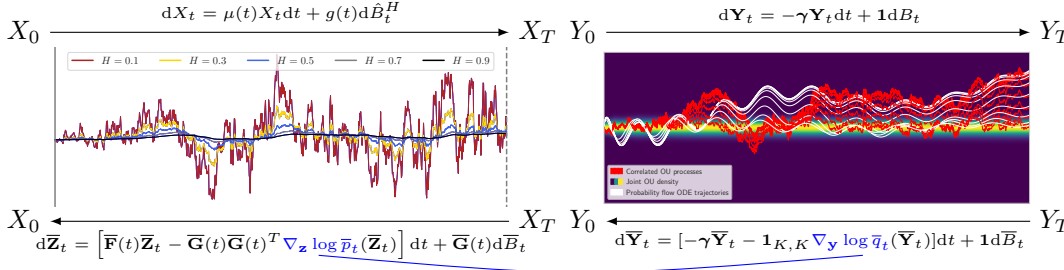

Figure 1: **Each data dimension transitions to a known prior distribution through a forward process that approximates a fractional diffusion process.** The Hurst index $H$ on the LHS interpolates between the roughness of a Brownian driven SDE and the underlying integration in PF ODEs. The driving noise process is a linear combination of the correlated processes on the RHS, all driven by the same Brownian motion. The score function of these augmenting processes is available in closed form and serves as guidance for the unknown score function.

audio samples [7, 8], 3D objects [9, 10] or textures [11], medical images [12], aerospace applications [13], and DNA sequence design [14, 15]. Despite these successes, modern score-based generative models formulated in continuous-time [16] face limitations due to their reliance on a simplistic driving noise, the Brownian motion (BM) [17, 18, 19]. As a light-tailed process, using BM can results in slow convergence rates and susceptibility to mode-collapse, especially with imbalanced data [20]. Additionally, its purely Markovian nature may also make it hard to capture the full complexity and richness of real-world data. All these attracted a number of attempts for involving different noise types [20, 21]. In this paper, we propose leveraging fractional noises, particularly the renowned non-Markovian fractional BM (fBM) [22, 23] to drive diffusion models. fBM extends BM to stationary increments with a more complex dependence structure, *i.e.*, long-range dependence vs. roughness/regularity controlled by a Hurst index, a measure of "mild" or "wild" randomness [24]. This all comes at the expense of computational challenges and intractability of inference, mostly stemming from its non-Markovian nature. Moreover, deriving a reverse-time model poses theoretical challenges, as fBM is not only non-Markovian but also not a semimartingale [25]. To overcome these limitations, we leverage recent works in Markov approximations of fBM (MA-fBM) [26, 27] and establish a framework for training continuous-time score-based generative models using an approximate fractional diffusion process, as well as generating samples from the corresponding tractable reverse process. Notably, our method maintains the same number of score model evaluations during both training and data generation, with only a minimal increase in computational load. Our contributions are:

- We derive the time-reversal of forward dynamics driven by a Markovian approximation of fractional Brownian motion in a way that the dimensionality of the unknown part of the score function matches that of the data.
- We derive an explicit formulae for the marginals of the conditional forward process via a continuous reparameterization trick.
- We introduce a novel augmented score matching loss for learning the score function in our generative fractional diffusion model, which can be minimized by a score model of data-dimension.

Our experimental evaluation validates our contributions, demonstrating the gains of correlated-noise with long-term memory, approximated by a combination of a number of Markov processes, where the amount of processes further control the diverstiy.

**Differentiation from existing work**. Yoon et al. [20] generalizes score-based generative models from an underlying BM to a driving Lévy process, a stochastic process with independent and stationary increments. A driving noise with correlated increments is not included in the framework of Yoon et al. [20]. Conceptually, every Lévy process is a semimartingale [28]. Since fBM is not a Lévy process, it is not included in the framework of Yoon et al. [20]. The closest work to ours is Tong et al. [29] constructing a neural-SDE based on correlated noise and using the neural SDE as a forward process of a score-based generative model. Our framework with exact reverse-time model is based on the integral representation of fBM derived in Harms and Stefanovits [26] and the optimal approximation coefficients of Daems et al. [27], while the fractional noise in [29] is sparsely approximated by a linear combination of independent standard normal random variables without exact reverse-time

model. Moreover, the framework of Tong et al. [29] is limited to $H > \frac{1}{3}$ and only compatible with the Euler-Maruyama sample schema [30] while our framework is up to numerical stability applicable for any $H \in (0, 1)$ and compatible with any suitable SDE or ODE solver. To the best of our knowledge, we are the first to build a framework for continuous-time score-based generative models that includes driving noise processes converging to non-Markovian processes with infinite quadratic variation.

## 2 Background

Modeling the distribution transforming process of a score-based generative model through stochastic differential equations (SDEs) [16] offers a unifying framework to generate data from an unknown probability distribution. Instead of injecting a finite number of fixed noise scales via a Markov chain, infinitely many noise scales tailored to the continuous dynamics of the Markov process $\mathbf{X} = (\mathbf{X}_t)_{t \in [0,T]}$ are utilized during the distribution transformation, offering considerable practical advantages over discrete time diffusion models [16]. The forward dynamics, transitioning from a data sample $\mathbf{X}_0 \sim p_0$ to a tractable noise sample $\mathbf{X}_T \sim p_T$ are specified by a continuous drift function $\mathbf{f}$ and a continuous diffusion coefficient $g$. These dynamics define a diffusion process that solves the SDE

$$d\mathbf{X}_t = \mathbf{f}(\mathbf{X}_t, t)dt + g(t)d\mathbf{B}_t, \quad \mathbf{X}_0 \sim p_0 \tag{1}$$

driven by a multivariate BM $\mathbf{B}$. To sample data from noise, a reverse-time model is needed that defines the backward transformation from the tractable noise distribution to the data distribution. Whenever $\mathbf{X} = (\mathbf{X}_t)_{t \in [0,T]}$ is a stochastic process and $g$ is a function on $[0, T]$, we write $\overline{\mathbf{X}}_t = \mathbf{X}_{T-t}$ for the reverse-time model and $\bar{g}(t) = g(T - t)$ for the reverse-time function. The marginal density of the stochastic process $\mathbf{X}$ at time $t$ is denoted by $p_t$ throughout this work[2]. Remarkably, an exact reverse-time model to the forward model in eq. (1) is given by the backward dynamics [31, 32, 33]

$$d\overline{\mathbf{X}}_t = \left[\overline{\mathbf{f}}(\overline{\mathbf{X}}_t, t) - \bar{g}^2(t)\nabla_{\mathbf{x}} \log \bar{p}_t(\overline{\mathbf{X}}_t)\right] dt + \bar{g}(t)d\overline{\mathbf{B}}_t, \quad \overline{\mathbf{X}}_0 = \mathbf{X}_T \sim p_T, \tag{2}$$

where the only unknown is the score function $\nabla_{\mathbf{x}} \log p_t$, inheriting the intractability from the unknown initial distribution $p_0$. In addition to the stochastic dynamics, the reverse-time model provides deterministic backward dynamics via an ordinary differential equation (ODE) by the so called probability flow ODE (PF ODE) [16]

$$d\bar{\mathbf{x}}_t = \left[\overline{\mathbf{f}}(\bar{\mathbf{x}}_t, t) - \frac{1}{2}\bar{g}^2(t)\nabla_{\mathbf{x}} \log \bar{p}_t(\bar{\mathbf{x}}_t, t)\right] dt, \quad \mathbf{x}_T \sim p_T. \tag{3}$$

Stochasticity is only injected into the system through the random initialization $\mathbf{x}_T \sim p_T$, implying a deterministic and bijective map from noise to data [16]. Conditioning the forward process on a data sample $\mathbf{x}_0 \sim p_0$ results for linear $\mathbf{f}(\cdot, t)$ in a tractable Gaussian forward process with conditional score function $\nabla_{\mathbf{x}} \log p_{0t}(\mathbf{x}|\mathbf{x}_0)$ in closed form. To approximate the exact reverse-time model, this tractable score function is used to train a time-dependent score model $S_{\boldsymbol{\theta}}$ via score matching [34, 35]. Upon training, any solver for SDEs or ODEs can be utilized to generate data from noise by simulating the stochastic or deterministic backward dynamics of the reverse-time model with $S_{\boldsymbol{\theta}} \approx \nabla_{\mathbf{x}} \log p$.

**Simulation error of the reverse-time model.** The two main sources of error when simulating the reverse-time model are the approximation error due to $S_{\boldsymbol{\theta}}$ only approximating $\nabla_{\mathbf{x}} \log p$, and the discretization error, which arises from transitioning from continuous-time to discrete steps. Simulating the PF ODE with the Euler method over $N \in \mathbb{N}$ equidistant time steps results in a global error of order $N^{-1}$ [36]. In contrast, the expected global error for simulating the SDE using the Euler-Maruyama method is of a lower order $N^{-\frac{1}{2}}$, indicating a larger error for the same number of steps [30, 36]. From this perspective it is reasonable that sampling from the PF ODE requires fewer steps. Yet, the source of qualitative differences between sampling from the ODE and the SDE [16] remains unclear.

**A pathwise perspective on sampling.** The roughness of a path can be measured by its Hölder exponent $0 < \delta \leq 1$ [37]. For example, BM as the integrator in the backward dynamics eq. (2) has $\delta$-Hölder continuous paths for any $0 < \delta < \frac{1}{2}$, whereas the integrator $t \mapsto t$ of the PF ODE eq. (3) can be regarded as a Hölder continuous path with exponent $\delta = 1$. Therefore, from a pathwise perspective, we move away from a rough path when we sample using the PF ODE. An unexplored topic in score-based generative models is the interpolation between the SDE and the PF ODE in terms

---

[2]See Appendix I for the notational conventions of this work.

of the Hölder exponent. It remains to be examined whether there is, to some extent, an optimal degree of Hölder continuity in between, or if an even rougher path with $\delta \ll \frac{1}{2}$ could yield an advantageous data generator.

The process that naturally arises from this line of thought is fBM [22, 23] with Hurst index $H \in (0, 1)$, where almost all paths are Hölder continuous for any exponent $\delta < H$, controlled by $H$. In terms of roughness, the Hurst index interpolates between the paths of Brownian driven SDEs and those of the underlying integration in PF ODEs, while also offering the potential for even rougher paths. Motivated by these observations, we define a novel score-based generative model with underlying dynamics that approximate a fractional diffusion process.

## 3 Fractional driving noise

Before describing the challenges in defining a score-based generative model with control over the roughness of the distribution transforming path, we introduce fBM. The literature distinguishes between "Type I" fBM and "Type II" fBM [38] having stationary and non-stationary increments, respectively. The type II fBM, also called Riemann-Liouville fBM, possesses smaller deviations from its mean, potentially an advantageous property for a driving noise of a score-based generative model, since large deviations of the sampling process to the data mean can lead to sample artifacts [39]. Here and in the experiments we focus on type II fBM. However, our theoretical framework generalizes to both types as detailed in Appendix A. The empirical study of a score-based generative model approximating a fractional diffusion process driven by type I fBM is dedicated to future work. We begin with the definition of Riemann-Liouville fBM [22], a generalization of BM permitting correlated increments.

**Definition 3.1** (Type II Fractional Brownian Motion [22]). *Let $B = (B_t)_{t \geq 0}$ be a standard Brownian Motion (BM) and $\Gamma$ the Gamma function. The centered Gaussian process*

$$B_t^H = \frac{1}{\Gamma(H + \frac{1}{2})} \int_0^t (t - s)^{H - \frac{1}{2}} \mathrm{d}B_s, \quad t \geq 0, \tag{4}$$

*uniquely characterized in law by its covariances*

$$\mathbb{E}\left[B_t^H B_s^H\right] = \frac{1}{\Gamma^2(H + \frac{1}{2})} \int_0^{\min\{t,s\}} ((t - u)(s - u))^{H - \frac{1}{2}} \mathrm{d}u, \quad t, s \in [0, \infty) \tag{5}$$

*is called type II fractional Brownian motion (fBM) with Hurst index $H \in (0, 1)$.*

BM being the unique continuous and centered Gaussian process with covariance $\min\{t, s\}$ is recovered for $H = 0.5$, since $\Gamma(1) = 1$. In comparison to the purely Brownian setting with independent increments (diffusion), the path of $B^H$ becomes more smooth for $H > 0.5$ due to positively correlated increments (super-diffusion) and more rough for $H < 0.5$ due to negatively correlated increments (sub-diffusion). These three regimes are reflected in the Hölder exponent of $\delta < H$ for almost all paths.

**Generalization challenges**. The most challenging part in defining a score-based generative model driven by fBM is the derivation of a reverse-time model. Due to its covariance structure, fBM is not a Markov process [40] and the shift in the roughness of the sample path leads to changes in its quadratic variation: from $t$ in the purely Brownian (diffusion) regime to zero in the smooth regime, and to infinite in the rough regime [30]. For that reason fBM is neither a Markov process nor a semimartingale [25] for all $H \neq 0.5$. Hence, we cannot make use of the Markov property or the Kolmogorov equations (Fokker-Planck) that are used to derive the reverse-time model of Brownian driven SDEs [31, 32, 33]. See Appendix H for a more illustrative view of the problem. The existence of a reverse-time model can be proven in the smooth regime of fBM [41]. However, due to the absence of an explicit score function in Darses and Saussereau [41] it does not provide a sufficient structure to train a score-based generative model.

To overcome this difficulty we follow [26, 27] and define the driving noise of our generative model by a linear combination of Markovian semimartingales. The approximation is based on the exact infinite-dimensional Markovian representation of fBM given in Theorem A.2.

**Definition 3.2** (Markov approximation of fBM [26, 27]). *Choose $K \in \mathbb{N}$ Ornstein–Uhlenbeck (OU) processes*

$$Y_t^k = \int_0^t e^{-\gamma_k(t-s)} \mathrm{d}B_s, \quad k \in \mathbb{N}, \quad t \geq 0, \tag{6}$$

*with speeds of mean reversion $\gamma_1, ..., \gamma_K$ and dynamics $\mathrm{d}Y_t^k = -\gamma_k Y_t^k \mathrm{d}t + \mathrm{d}B_t$. Given a Hurst index $H \in (0,1)$ and a geometrically spaced grid $\gamma_k = r^{k-n}$ with $r > 1$ and $n = \frac{K+1}{2}$ we call the process*

$$\hat{B}_t^H := \sum_{k=1}^{K} \omega_k Y_t^k, \quad H \in (0,1), \quad t \geq 0, \tag{7}$$

*Markov-approximate fractional Brownian motion (MA-fBM) with approximation coefficients $\omega_1, ..., \omega_K \in \mathbb{R}$ and denote by $\hat{\mathbf{B}}^H = (\hat{B}_1^H, ..., \hat{B}_D^H)$ the corresponding $D$-dimensional process where $\hat{B}_i^H$ and $\hat{B}_j^H$ are independent for $i \neq j$ inheriting independence from the underlying standard BMs $B_i$ and $B_j$.*

Our framework is conceptually independent of the specific choice of spatial grid and approximation coefficients. To achieve strong convergence rates with a high polynomial order in $K$ for $H < 0.5$ in the driving noise to fBM, one may follow the approach outlined in Harms [42]. Consequently, our framework includes driving noise processes that converge to non-Markovian processes with infinite quadratic variation. For computational efficiency, we instead follow the approach of Daems et al. [27] to choose the $L^2(\mathbb{P})$ optimal approximation coefficients for a given $K$, achieving empirically good results in approximating fBM, even with a small number of OU processes.

**Proposition 3.3** (Optimal Approximation Coefficients [27]). *The optimal approximation coefficients $\boldsymbol{\omega} = (\omega_1, ..., \omega_K) \in \mathbb{R}^K$ for a given Hurst index $H \in (0,1)$, a terminal time $T > 0$ and a fixed geometrically spaced grid to minimize the $L^2(\mathbb{P})$-error*

$$\mathcal{E}(\boldsymbol{\omega}) := \int_0^T \mathbb{E}\left[\left(B_t^H - \hat{B}_t^H\right)^2\right] \mathrm{d}t \tag{8}$$

*are given by the closed-form expression $\boldsymbol{A}\boldsymbol{\omega} = \boldsymbol{b}$ with*

$$\boldsymbol{A}_{i,j} := \frac{2T + \frac{e^{-(\gamma_i + \gamma_j)T} - 1}{\gamma_i + \gamma_j}}{\gamma_i + \gamma_j}, \quad \boldsymbol{b}_k := \frac{T}{\gamma_k^{H+\frac{1}{2}}} P\left(H + \frac{1}{2}, \gamma_k T\right) - \frac{H + \frac{1}{2}}{\gamma_k^{H+\frac{3}{2}}} P\left(H + \frac{3}{2}, \gamma_k T\right) \tag{9}$$

*and where $P(z,x) = \frac{1}{\Gamma(z)} \int_0^x t^{z-1} e^{-t} \mathrm{d}t$ is the regularized lower incomplete gamma function.*

MA-fBM serves as the driving noise of our generative model, replacing BM in the distribution transforming process solving eq. (1), approximating a fractional diffusion process. See Figure 1 for an illustration of the underlying processes.

## 4 A score-based generative model based on fractional noise

In this section, we define a continuous-time score-based generative model driven by MA-fBM. A detailed treatment of the theory can be found in Appendix A. We begin with the forward dynamics, transitioning data to noise.

**Definition 4.1** (Forward process). *Let $\hat{\mathbf{B}}^H$ be a $D$-dimensional MA-fBM with Hurst index $H \in (0,1)$. For continuous functions $\mu : [0,T] \rightarrow \mathbb{R}$ and $g : [0,T] \rightarrow \mathbb{R}$ we define the forward process $\mathbf{X} = (\mathbf{X}_t)_{t \in [0,T]}$ of a generative fractional diffusion model (GFDM) by*

$$\mathrm{d}\mathbf{X}_t = \mu(t)\mathbf{X}_t \mathrm{d}t + g(t)\mathrm{d}\hat{\mathbf{B}}_t^H, \quad \mathbf{X}_0 = \mathbf{x}_0 \sim p_0, \quad t \in [0,T], \tag{10}$$

*where $p_0$ is the unknown data distribution from which we aim to sample from.*

Considering both the forward process as well as the OU processes defining the driving noise $\hat{\mathbf{B}}^H$, we have for every data dimension an augmented vector of correlated processes $(X, Y^1, \ldots, Y^K)$, driven by the same BM, approximating the time-correlated behavior of a one-dimensional fractional diffusion process [27]. We denote the stacked process of the $D$ augmented vectors as $\mathbf{Z} = (\mathbf{Z}_t)_{t \in [0,T]}$ and refer to the resulting $D(K+1)$-dimensional process as the *augmented forward process*. Rewriting the dynamics of the forward process we observe that the augmented forward process $\mathbf{Z}$ solves a linear SDE. Hence, $\mathbf{Z}|\mathbf{x}_0$, the augmented forward process conditioned on a data sample $\mathbf{x}_0 \sim p_0$, is a linear transformation of BM. Thus $\mathbf{Z}|\mathbf{x}_0$ is a Gaussian process and so is $\mathbf{X}|\mathbf{x}_0$ [43]. For each dimension $1 \leq d \leq D$, we have a system of $K+1$ trajectories that transform $\mathbf{x}_{0,d}$ according to the augmented forward process with $D = 1$, following the dynamics

$$\mathrm{d}\mathbf{Z}_t = \mathbf{F}(t)\mathbf{Z}_t \mathrm{d}t + \mathbf{G}(t)\mathrm{d}B_t, \tag{11}$$

where all $K + 1$ processes are driven by the same one-dimensional BM $B$ with matrix valued functions $\mathbf{F}$ and $\mathbf{G}$ defined in Appendix A.2. To efficiently sample for every $t \in (0, T]$ from the conditional augmented forward distribution during training, we characterize its marginal statistics.

**Derivation of marginal statistics.** The marginal mean $\mathbb{E}[\mathbf{X}_t | \mathbf{x}_0] = \mathbf{x}_0 \exp(\int_0^t \mu(s) \mathrm{d}s)$ of the conditional forward process is unaffected by changing the driving noise to MA-fBM, and the mean of the augmenting OU processes is zero. See Appendix A.2 for a detailed derivation of the marginal statistics of the augmenting processes. The missing components in the marginal covariance matrix $\mathbf{\Sigma}_t$ of the conditional augmented forward process $\mathbf{Z} | \mathbf{x}_0$ are the marginal variance of the forward process and the marginal correlation between the conditional forward process and the augmenting processes. We derive by reparameteriziation an explicit formula for the marginal variance of the conditional forward process. This generalizes the formula for the perturbation kernel $p_{0t}(\mathbf{x} | \mathbf{x}_0) = \mathcal{N}(\mathbf{x}; c(t)\mathbf{x}_0, c^2(t)\sigma^2(t)\mathbf{I}_D)$ given in Karras et al. [44] to a driving MA-fBM and is reminiscent of the reparameterization trick used in discrete time.

**Proposition 4.2** (Continuous Reparameterization Trick). *The forward process $\mathbf{X}$ of GFDM conditioned on $\mathbf{x}_0 \in \mathbb{R}^D$ admits the continuous reparameterization*

$$\mathbf{X}_t = c(t)\left(\mathbf{x}_0 + \int_0^t \alpha(t, s)\mathrm{d}\mathbf{B}_s\right) \sim \mathcal{N}(c(t)\mathbf{x}_0, c^2(t)\sigma^2(t)\mathbf{I}_D) \tag{12}$$

*with $c(t) = \exp\left(\int_0^t \mu(s)\mathrm{d}s\right)$ and $\sigma^2(t) = \int_0^t \alpha^2(t, s)\mathrm{d}s$ where $\alpha$ is given by*

$$\alpha(t, s) = \sum_{k=1}^K \omega_k \left[\frac{g(s)}{c(s)} - \gamma_k \int_s^t f_k(u, s)\mathrm{d}u\right], \quad f_k(u, s) = \frac{g(u)}{c(u)}e^{-\gamma_k(u-s)}. \tag{13}$$

*Sketch of Proof.* Reparameterization of the forward dynamics in eq. (10) and the Stochastic Fubini Theorem yields the Gaussian process $\mathbf{X}_t = c(t)(\mathbf{x}_0 + \int_0^t \alpha(t, s)\mathrm{d}\mathbf{B}_s)$ with variance $\mathbb{V}[\mathbf{X}_t] = c^2(t) \int_0^t \alpha^2(t, s)\mathrm{d}s$ by Itô isometry. See Theorem A.3 for the full proof. $\qquad \square$

By the above definition of $\alpha$, we retrieve the perturbation kernel of the purely Brownian setting given in Karras et al. [44, Equation 12] for $K = 1$, $\gamma_1 = 0$ and $\omega_1 = 1$. When, depending on the choice of forward dynamics, $\int_0^t \alpha(t, s)\mathrm{d}s$ is not accessible in closed form, $\mathbf{\Sigma}_t$ can be described by an ODE and solved numerically as described in Appendix B. Thus our method admits any choice of forward dynamics in terms of $\mu$ and $g$.

**Explicit fractional forward dynamics.** Although our framework is not bound to any specific dynamics, this work's empirical evaluation focuses on *Fractional Variance Exploding* (FVE) dynamics given by

$$\mathrm{d}\mathbf{X}_t = \sigma_{min}\left(\frac{\sigma_{max}}{\sigma_{min}}\right)^t \sqrt{2 \log \frac{\sigma_{max}}{\sigma_{min}}} \mathrm{d}\hat{\mathbf{B}}_t^H, \quad t \in [0, T] \tag{14}$$

with $(\sigma_{min}, \sigma_{max}) = (0.01, 50)$ and *Fractional Variance Preserving* (FVP) dynamics given by

$$\mathrm{d}\mathbf{X}_t = -\frac{1}{2}\beta(t)\mathbf{X}_t\mathrm{d}t + \sqrt{\beta(t)}\mathrm{d}\hat{\mathbf{B}}_t^H, \quad t \in [0, T] \tag{15}$$

with $\beta(t) = \beta(t) = \bar{\beta}_{\min} + t\left(\bar{\beta}_{max} - \bar{\beta}_{min}\right)$ and $(\bar{\beta}_{min}, \bar{\beta}_{max}) = (0.1, 20)$ [16]. Leveraging the continuous reparameterization trick we derive in Appendix B the conditional marginal covariance matrix of FVE in closed form. To the best of our knowledge, the integral in eq. (13), needed to compute $\alpha$ in the setting of FVP dynamics, is not accessible in closed form. Therefore, we use a numerical ODE solver to estimate this quantity for FVP dynamics. See Appendix B for details on the computation of the marginal variances and an illustration of the resulting variance schedules.

**The reverse-time model.** We observe that the augmented forward dynamics of GFDM are already encompassed in the general framework presented in Song et al. [16, Appendix A], although they differ from the Variance Exploding (VE), Variance Preserving (VP), and sub-VP dynamics discussed therein. To simplify notation, we use $p_t$ here to denote the marginal density of both $\mathbf{Z}_t$ and $\mathbf{X}_t$. The specific density referred to will be clear from the context. By the significant results of [31, 32, 33], the reverse-time model of GFDM is given by the backward dynamics

$$\mathrm{d}\overline{\mathbf{Z}}_t = \left[\overline{\mathbf{F}}(t)\overline{\mathbf{Z}}_t - \overline{\mathbf{G}}(t)\overline{\mathbf{G}}(t)^T \nabla_{\mathbf{z}} \log \overline{p}_t(\overline{\mathbf{Z}}_t)\right]\mathrm{d}t + \overline{\mathbf{G}}(t)\mathrm{d}\overline{\mathbf{B}}_t, \quad t \in [0, T]. \tag{16}$$

However, a direct application of [16] would require to train a score model with input and output dimension of $D(K + 1)$. By proposing *augmented score matching* below, we show that learning a score model with input and output dimension $D$ is sufficient, enabling the use of the same highly curated model architecture as in traditional diffusion models to approximate the score function.

**Augmented score matching.** We condition the score function $\nabla_{\mathbf{z}} \log p_t$ on a data sample $\mathbf{x}_0 \sim p_0$ *and additionally* on the states of the stacked vector $\mathbf{Y}_t^{[K]} := (\mathbf{Y}_t^1, ..., \mathbf{Y}_t^K)$ of augmenting processes. To train our time-dependent score model $s_{\boldsymbol{\theta}}$ we propose the *augmented score matching loss*

$$\mathcal{L}(\boldsymbol{\theta}) := \mathbb{E}_t \left\{ \mathbb{E}_{(\mathbf{X}_0, \mathbf{Y}_t^{[K]})} \mathbb{E}_{(\mathbf{X}_t | \mathbf{Y}_t^{[K]}, \mathbf{X}_0)} \left[ \| s_{\boldsymbol{\theta}}(\mathbf{X}_t - \sum_k \eta_t^k \mathbf{Y}_t^k, t) - \nabla_{\mathbf{x}} \log p_{0t}(\mathbf{X}_t | \mathbf{Y}_t^{[K]}, \mathbf{X}_0) \|_2^2 \right] \right\}. \tag{17}$$

The weights $\eta_t^1, ..., \eta_t^K$ arise from conditioning $\mathbf{Z}_t | \mathbf{x}_0$ on $\mathbf{Y}_t^{[K]}$ and the time points $t$ are uniformly sampled from $\mathcal{U}[0, T]$. We show in the following that the optimal $s_{\boldsymbol{\theta}}$ w.r.t. the *augmented score matching loss* is the $L^2$-optimal approximation of the score function of our reverse-time model.

**Proposition 4.3** (Optimal Score Model). *Assume that $s_{\boldsymbol{\theta}}$ is optimal w.r.t. the augmented score matching loss $\mathcal{L}$. The score model*

$$S_{\theta}(\mathbf{Z}_t, t) := \left( s_{\boldsymbol{\theta}}(\mathbf{X}_t - \sum_k \eta_t^k \mathbf{Y}_t^k, t), -\eta_t^1 s_{\boldsymbol{\theta}}(\mathbf{X}_t - \sum_k \eta_t^k \mathbf{Y}_t^k, t), ..., -\eta_t^K s_{\boldsymbol{\theta}}(\mathbf{X}_t - \sum_k \eta \mathbf{Y}_t^k, t) \right) \tag{18}$$

*yields the optimal $L^2(\mathbb{P})$ approximation of $\nabla_{\mathbf{z}} \log p_t(\mathbf{Z}_t)$ via*

$$S_{\boldsymbol{\theta}}(\mathbf{Z}_t, t) + \nabla_{\mathbf{z}} \log q_t(\mathbf{Y}_t^{[K]}) \approx \nabla_{\mathbf{z}} \log p_t(\mathbf{Z}_t). \tag{19}$$

*Sketch of Proof.* Using the relation $\nabla_{\mathbf{x}} \log p_{0t} = -\eta_t^k \nabla_{\mathbf{y}^k} \log p_{0t}$ and the independence of $\mathbf{X}_0$ and $\mathbf{Y}_t^{[K]}$ yields the claim. See Appendix A.3 for the full proof. $\square$

In addition to the result that a score model of data dimension $D$ minimizes the proposed *augmented score matching loss*, Proposition 4.3 also implies that GFDM requires the same number of score model evaluations during sampling from the reverse-time model as traditional diffusion models. This is because, for a given time point $t$, we only need to evaluate $s_{\boldsymbol{\theta}}(\cdot, t)$ once at $\mathbf{X}_t - \sum_k \eta_t^k \mathbf{Y}_t^k$ to compute $S_{\boldsymbol{\theta}}(\mathbf{Z}_t, t)$ according to eq. (18), and $S_{\boldsymbol{\theta}}$ is all that is required to approximate the reverse-time dynamics described below. We provide a thorough quantitative evaluation of compute time in seconds for GFDM in Appendix F, validating the theoretical reasoning in this section that GFDM incur only minimal additional computational cost.

**Sampling from reverse-time model.** Once we trained our score model $S_{\boldsymbol{\theta}}$ via augmented score matching, we simulate the reverse-time model backward in time and sample from the reverse-time model via the SDE

$$d\overline{\mathbf{Z}}_t = \left\{ \overline{\mathbf{F}}(t) \overline{\mathbf{Z}}_t - \overline{\mathbf{G}}(t) \overline{\mathbf{G}}(t)^T \left[ \overline{S}_{\boldsymbol{\theta}}(\overline{\mathbf{Z}}_t, t) + \nabla_{\mathbf{z}} \log \overline{q}_t(\overline{\mathbf{Y}}_t^{[K]}) \right] \right\} dt + \overline{\mathbf{G}}(t) d\overline{\mathbf{B}}_t, \quad t \in [0, T] \tag{20}$$

or the corresponding augmented PF ODE [16]

$$d\overline{\mathbf{z}}_t = \left\{ \overline{\mathbf{F}}(t) \overline{\mathbf{z}}_t - \frac{1}{2} \overline{\mathbf{G}}(t) \overline{\mathbf{G}}(t)^T \left[ \overline{S}_{\boldsymbol{\theta}}(\overline{\mathbf{z}}_t, t) + \nabla_{\mathbf{z}} \log \overline{q}_t(\overline{\mathbf{y}}_t^{[K]}) \right] \right\} dt, \quad t \in [0, T], \tag{21}$$

where we initialize in both cases the reverse dynamics with the centered (non-isotropic) Gaussian $\overline{\mathbf{Z}}_0$ with covariance matrix $\boldsymbol{\Sigma}_T$. To traverse backward from noise to data, we may deploy any suitable SDE or ODE solver. In both cases, for each data dimension, we have $K + 1$ trajectories that transform the Gaussian initialization into an approximate sample of the data distribution. The PF ODE enables in addition negative log-likelihoods (NLLs) estimation of test data under the learned density [16]. See Appendix G for the computation details of NLLs.

**Remark 4.4.** *We showed in this section that it suffices to approximate a $D$-dimensional score to reverse the $D(K + 1)$-dimensional MA-fBM driven SDE with unknown starting distribution. Since this holds for any fixed $K \in \mathbb{N}$ an interesting task is to examine the behaviour of the reverse-time model as $K \to \infty$ and potentially link it to the dynamics of a reverse-time model of true fBM. To the best of our knowledge, existence of such a reverse-time model is not known for $H < 0.5$ and the drift of the reverse-time model for $H > 0.5$ lacks sufficient structure to train a score-based generative model [41].*

| **FVE**$(H=0.5)$ | FID $\downarrow$ | NLLs Test $\downarrow$ | VS$_p$ $\uparrow$ | **FVP**$(H=0.5)$ | FID $\downarrow$ | NLLs Test $\downarrow$ | VS$_p$ $\uparrow$ |
|---|---|---|---|---|---|---|---|
| VE (retrained) | 10.82 | 2.73 | 24.20 | VP (retrained) | **1.44** | **2.38** | 23.64 |
| $K=1$ | 10.30 | **2.55** | 24.22 | $K=1$ | 2.81 | 3.90 | 23.69 |
| $K=2$ | 9.89 | 3.03 | 24.15 | $K=2$ | 2.92 | 4.57 | 23.63 |
| $K=3$ | **9.74** | 2.93 | 24.42 | $K=3$ | 3.51 | 7.02 | 23.78 |
| $K=4$ | 11.25 | 3.10 | **24.54** | $K=4$ | 1.86 | 5.71 | 24.50 |
| $K=5$ | 25.51 | 3.94 | 23.08 | $K=5$ | 4.89 | 7.09 | **24.56** |

Table 1: Effect of augmenting processes on conditional image generation on MNIST for FVE and FVP dynamics.

| **MNIST** | $H=0.9$ | | $H=0.7$ | | $H=0.5$ | | $H=0.1$ | | **CIFAR10** | $H=0.9$ | | $H=0.7$ | | $H=0.5$ | | $H=0.1$ | |
|---|---|---|---|---|---|---|---|---|---|---|---|---|---|---|---|---|---|
| | FID $\downarrow$ | VS$_p$ $\uparrow$ | FID $\downarrow$ | VS$_p$ $\uparrow$ | FID $\downarrow$ | VS$_p$ $\uparrow$ | FID $\downarrow$ | VS$_p$ $\uparrow$ | | FID $\downarrow$ | VS$_p$ $\uparrow$ | FID $\downarrow$ | VS$_p$ $\uparrow$ | FID $\downarrow$ | VS$_p$ $\uparrow$ | FID $\downarrow$ | VS$_p$ $\uparrow$ |
| **BM driven** | | | | | | | | | **BM driven** | | | | | | | | |
| VE (retrained) | - | - | - | - | 10.82 | 24.20 | - | - | VE (retrained) | - | - | - | - | 5.20 | 3.42 | - | - |
| VP (retrained) | - | - | - | - | 1.44 | 23.64 | - | - | VP (retrained) | - | - | - | - | 4.85 | 3.28 | - | - |
| **MA-fBM driven** | | | | | | | | | **MA-fBM driven** | | | | | | | | |
| FVP$(H,K=1)$ | 2.86 | 23.56 | 3.01 | 23.78 | 2.81 | 23.69 | 2.92 | 23.59 | FVP$(H,K=1)$ | **4.79** | **3.53** | 4.96 | **3.84** | 4.19 | **3.99** | 4.60 | **3.46** |
| FVP$(H,K=2)$ | 1.93 | 24.00 | 2.30 | 23.82 | 2.92 | 23.63 | 2.56 | 23.82 | FVP$(H,K=2)$ | 3.77 | **3.60** | 4.17 | 3.35 | 4.85 | **4.04** | 5.77 | **3.43** |
| FVP$(H,K=3)$ | 0.72 | 24.18 | 2.67 | 23.96 | 3.51 | 23.78 | 4.87 | 23.60 | FVP$(H,K=3)$ | 14.22 | 3.38 | 6.12 | 3.39 | 6.32 | **3.49** | 5.95 | 3.34 |
| FVP$(H,K=4)$ | **1.22** | **24.76** | **0.86** | **24.39** | 1.86 | **24.50** | 6.25 | 23.89 | FVP$(H,K=4)$ | 29.72 | **3.67** | 8.35 | 3.24 | 8.85 | **3.65** | 5.02 | 3.26 |
| FVP$(H,K=5)$ | 2.17 | **25.15** | 1.36 | 24.63 | 4.89 | **24.56** | 9.57 | 23.70 | FVP$(H,K=5)$ | 69.06 | **6.61** | 35.91 | 5.20 | 96.54 | **7.30** | 7.38 | 3.11 |

| (a) Conditional image generation on MNIST. | (b) Conditional image generation on CIFAR10. |
|---|---|

Table 2: FID and pixel-wise diversity VS$_p$ of GFDM compared to the original setting of purely Brownian driven VE and VP. In bold the scores that are better than both purely Brownian driven dynamics. The overall best scores within the experiment are boxed in, indicating that the highest scores on both datasets are achieved in the super-diffusive regime for $H=0.9$.

# 5 Experiments

We conduct experiments on MNIST and CIFAR10 to evaluate the ability of GFDM to generate real images. First, we measure the quality and the pixel-wise diversity of the generated images across different numbers of augmenting processes and various Hurst indices, showing that the super-diffusive regime with $H > 0.5$ yields better performance compared to the purely Brownian driven dynamics. Second, we further evaluate the best performing models in terms of class-wise image quality and class-wise distribution coverage. We measure image quality by the Frechét Inception Distance (FID) [45] and the Inception score (IS) [46], pixel-wise diversity by the pixel Vendi Score (VS$_p$) [47] and class-wise distribution coverage by improved recall (Recall) [48]. See Appendix D for the implementation details and additional experimental results. We begin with the empirical evaluation of how the augmenting processes affect performance on MNIST.

**Effect of augmentation on MNIST**. To isolate the effect of the augmenting processes on MNIST while minimally adapting the driving noise distribution, we fix $H = 0.5$ so that the weighted sum of the augmenting processes approximates BM, rather than fBM. We observe an increase of the pixel-wise diversity VS$_p$ for both FVE and FVP dynamics, with increasing $K$. In Table 1 we can observe that VS$_p$ increases from $24.20$ to $24.54$ for FVE dynamics and from $23.64$ to $24.56$ for FVP dynamics. The enhanced pixel-wise diversity on MNIST comes at the cost of a reduced likelihood of test data under the learned density, indicated by a higher NLLs for more augmenting processes.

**Quality results across different Hurst indices**. On both, MNIST and CIFAR10, we obtain the best performance in terms of FID and VS$_p$ in the super-diffusive regime with $H = 0.9$ and FVP dynamics. On MNIST we achieve state of the art FID of $0.72$, compared to an FID of $1.44$ with the purely Brownian VP dynamics (Table 2a). Comparing FVP to the best-performing purely BM driven VP dynamics, we observe not only an improvement in quality but also an increase in pixel-wise diversity from $23.64$ to $24.18$, as measured by VS$_p$ . In Table 2b we observe the same behaviour on CIFAR10. The best performing configuration in terms of FID and pixel-wise diversity is achieved for FVP$(H = 0.9, K = 2)$ with an FID of $3.77$ instead of $4.85$ and an VS$_p$ of $3.60$ instead of $3.28$. Additionally, in Figure 7, we show the FID evolution of the super-diffusive regime

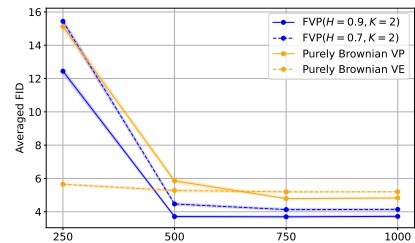

Figure 2: Comparison of the super-diffusive regime and purely Brownian dynamics in terms of average FID over three rounds of sampling plotted across different NFEs.

for various numbers of augmenting processes, showing a similar pattern that either that $K = 2$ or

| Metric | Dynamics | airplane | automobile | bird | cat | deer | dog | frog | horse | ship | truck |
|---|---|---|---|---|---|---|---|---|---|---|---|
| FID ↓ | VP | 15.29 | 12.06 | 14.08 | 18.08 | 10.68 | 16.92 | 16.48 | 12.49 | 10.74 | 10.57 |
| | FVP($H = 0.7, K = 2$) | 14.67 | 9.55 | **14.02** | 16.97 | 11.05 | 17.14 | 16.43 | 10.97 | **9.91** | 8.81 |
| | FVP($H = 0.9, K = 2$) | **14.37** | **8.94** | 14.18 | **16.38** | **10.52** | **16.76** | **15.37** | **10.28** | 10.04 | **8.76** |
| Recall ↑ | VP | 0.6814 | 0.6186 | 0.6860 | 0.6466 | 0.7002 | 0.6730 | 0.6758 | 0.6392 | 0.6468 | 0.5982 |
| | FVP($H = 0.7 K = 2$) | 0.6838 | 0.6436 | 0.6870 | 0.6712 | 0.7140 | 0.6844 | 0.6922 | 0.6764 | 0.6550 | 0.6508 |
| | FVP($H = 0.9, K = 2$) | **0.7038** | **0.6614** | **0.7188** | **0.6842** | **0.7284** | **0.7096** | **0.7104** | **0.6806** | **0.6772** | **0.6852** |

Table 3: The class-wise image quality and class-wise distribution coverage of the super-diffusive regime FVP($H = 0.9, K = 2$) compared to the purely Brownian VP dynamics.

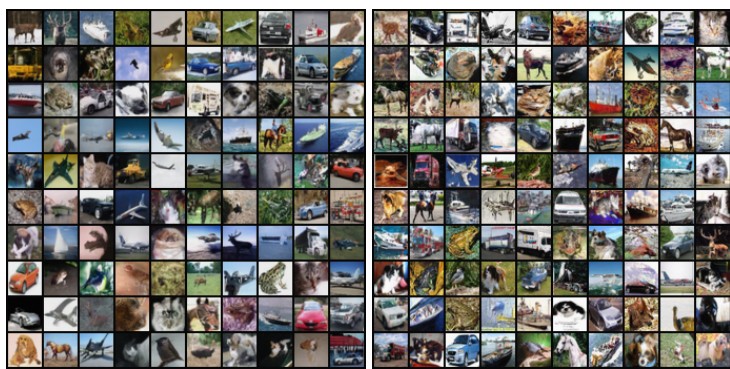

Figure 4: Visual comparison of PF ODE samples. (LHS) Purely Brownian VP dynamics. (RHS) Super-diffusive regime FVP($H = 0.9, K = 2$).

$K = 3$ yields the best performance across different datasets and dynamics. Evaluating the performance with different number of sampling steps in Figure 2 shows that the super-diffusive regime with $K = 2$ saturates already at 500 number of function evaluations (NFEs) on a lower level than both purely Brownian driven dynamics VP and VE. See Figure 2 in Appendix D for the exact FID values.

**Class-wise distribution coverage**. We evaluate the capability to generate samples from different classes in terms of FID and class-wise distribution coverage, measured by Recall, comparing the best-performing purely Brownian driven dynamics to the super-diffusive regime with $K = 2$. In Table 3 we observe that the super-diffusive regime with $K = 2$ outperforms in both FID and Recall, where $H = 0.7$ and $H = 0.9$ achieve better class-wise FID for all but two and one class, respectively (deer and dog for $H = 0.7$, bird for $H = 0.9$). Additionally, the super-diffusive regime shows improved class-wise distribution coverage, as indicated by a higher Recall across all classes. Overall, both $H = 0.7$ and $H = 0.9$ perform significantly better in terms of distribution coverage than VP dynamics, $H = 0.9$ being the best performing model.

**Sampling with the augmented probability flow ODE**. We compare the performance of sampling via the PF ODE for the best performing models from above. For MA-fBM driven dynamics, we have $K + 1$ deterministic trajectories for each pixel, traversing from noise to data. As shown in Figure 3, the PF ODE associated with purely Brownian dynamics outperforms the super-diffusive regime in terms of FID, while the super-diffusive regime achieves the overall highest pixel-wise diversity of $VS_p = 4.89$ confirmed mildly perceptually in Figure 4. See Appendix E for additional visualization of the generated data.

|  | FID ↓ | IS ↑ | $VS_p$ ↑ |
|---|---|---|---|
| **Sampled with SDE** | | | |
| VE (retrained) | 5.20 | 9.60 | 3.42 |
| VP (retrained) | 4.85 | 9.64 | 3.28 |
| FVP($H = 0.7, K = 2$) | 4.17 | 9.51 | 3.35 |
| FVP($H = 0.9, K = 2$) | **3.77** | 9.41 | 3.60 |
| **Sampled with PF ODE** | | | |
| VE (retrained) | 6.40 | 9.22 | 3.14 |
| VP (retrained) | 5.63 | 9.23 | 3.91 |
| FVP($H = 0.7, K = 2$) | 12.23 | **9.73** | 4.38 |
| FVP($H = 0.9, K = 2$) | 12.26 | 9.55 | **4.89** |

Figure 3: Quantitative performance comparison of SDE and PF ODE sampling.

Our experiments show that, compared to purely Brownian dynamics, the super-diffusive regime of MA-fBM yields higher image quality with fewer NFEs, improved pixel-wise diversity and better distribution coverage.

# 6 Related work

**Diffusion models in continuous-time**. The seminal work of Song et al. [16] offers a unifying framework modeling the distribution transforming process by a stochastic processes in continuous-time with exact reverse-time model. Extensive research has been carried out to examine [44, 49, 50] and extend [39, 51, 52, 53, 54, 55] the continuous-time view on generative models through the lens of

SDEs, including deterministic corruptions [56] and blurring diffusion [57]. While critic on this view question the usefulness of the theoretical superstructure [58], others extend in line with our work the theoretical framework to new types of underlying diffusion processes [59]. Conceptually similar to our work, Yoon et al. [20] generalizes the score-based generative model from an underlying Brownian motion to a driving Lévy process, thereby dropping the Gaussian assumptions on the increments. In contrast to our work, the framework of Yoon et al. [20] does not include correlated increments. Importantly, every Lévy process is a semimartingale, which means that fBM is not a Lévy process.

**Fractional noises in machine learning**. Recently, Hayashi and Nakagawa [60] considered neural-SDEs driven by fractional noise. Yet they do not study diffusion models. The closest work to our work, Tong et al. [29] approximated the type-II fBM with sparse Gaussian processes constructing a neural SDE as a forward process of a score-based generative model, without exact reverse-time dynamics. Unfortunately, they are also limited to Euler-Maruyama solvers and to the case of $H > 1/3$, while our framework is up to numerical stability applicable for any $H \in (0, 1)$ and compatible with any suitable SDE or ODE solver. Daems *et al.* [27], who inspired our Markov-approximate noise, includes a more elaborate discussion as well as a variational inference framework for MA-fBM.

**Rough path theory**. The pathwise analysis of SDEs driven by processes with a Hölder exponent less than $0.5$, including fBM for $H < 0.5$ and BM, is encompassed by rough path theory [37]. Rough path theory is applied in machine learning in several ways including (i) deriving stability bounds for the trained weights of a residual neural network [61], (ii) enabling rough control of neural ODEs [62], and (iii) modeling long time series behavior via neural rough differential equations [63, 64]. In finance the famous Black-Scholes model [65] is driven by BM, while more recent continuous-time models employ fractional noise to model price processes [66, 67] or rough volatility [68, 69] to more closely mimic real-world behavior.

# 7 Conclusion

In this work, we propose a generalized framework of continuous-time score-based generative models, introducing a novel generative model driven by MA-fBM with control over the roughness of distribution transformation paths via augmenting processes. Despite the increased dimensionality of the forward process, learning a score model with the dimensionality of the data distribution, guided by the marginal known score of the augmenting processes, is sufficient. Consequently, both training and sampling is efficient. Our experimental results show that the super-diffusive regime of our MA-fBM driven dynamics achieves superior performance in terms of FID and pixel-wise diversity. Additionally, the FID saturates at a lower level with fewer function evaluations compared to purely Brownian driven dynamics. The super-diffusive regime also improves class-wise distribution coverage, as measured by Recall. Based on these results, GFDM offers a promising alternative to traditional diffusion models for generating data from an unknown distribution.

**Limitations and future work**. Several practical and theoretical questions remain open. While we draw our conclusions from experiments conducted on MNIST and CIFAR10, generalizing the observed behavior to other datasets and data modalities may not be valid. In future work, we aim to empirically and theoretically determine the optimal degree of correlated noise, and thus the optimal Hurst index, for training and sampling across different data modalities. Beyond image data, a particularly interesting modality could be the generation of rough time series data using dynamics of the sub-diffusive regime. A theoretical open question is the limiting behavior of GFDM's reverse dynamics with infinitely many augmenting processes and whether this limit is connected to the reverse time model for true fBM. An intriguing extension would be to adapt the dynamics of our framework to switch between two unknown distributions. This adaptation would enable the use of MA-fBM driven dynamics in the sciences to model real-world evolution between two states of unknown distributions. This is a promising direction, as the assumption of independent increments in real-world noise processes is often too strong.

**Broader impact**. Our contribution advances generative modeling by introducing a specific driving noise process to improve the learning of an unknown distribution. This conceptual work aims to support impactful applications of generative modeling, such as molecular structure generation, medical imaging, drug discovery, and DNA sequence design. However, we acknowledge that generative models can reflect biases in the datasets they are trained on and may pose risks, including misuse for human impersonation and the spread of fake content.

## Acknowledgements

We would like to give a special thanks to Thorsten Selinger for his support in utilizing Fraunhofer HHI's GPU cluster. We also thank the anonymous reviewers for their constructive feedback, which helped improve our work. This work was supported by the Federal Ministry of Education and Research (BMBF) as grants [SyReal (01IS21069B)]. R.M-S. & M.A. acknowledge funding from the *QuantIC* Project funded by EPSRC Quantum Technology Programme (grant EP/MO1326X/1, EP/T00097X/1), and *dotPhoton AG*. R.M-S acknowledges funding from EP/R018634/1, EP/T021020/1, EP/Y029178/1, and *Google*. SN acknowledges funding from the German Federal Ministry of Education and Research under the grant BIFOLD24B. RD acknowledges funding from the Flemish Government under the "Onderzoeksprogramma Artificiële Intelligentie (AI) Vlaanderen" programme and from Flanders Make under the SBO project CADAIVISION. TB was supported by a UKRI Future Leaders Fellowship [grant number MR/Y018818/1]. MO has been partially funded by Deutsche Forschungsgemeinschaft (DFG) - Project - ID 318763901 - SFB1294. WS acknowledges financial support by the German Research Foundation (DFG) - Research Unit KI-FOR 5363 (project ID: 459422098).

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

## Appendix

# A  The mathematical framework of generative fractional diffusion models

In this section we provide the mathematical details of the score-based generative model defined in the main paper. The driving noise of the underlying stochastic process is based on the affine representation of fractional processes from Harms and Stefanovits [26] and further simplified by the closed form expression to determine optimal approximation coefficients of Daems et al. [27].

## A.1  A Markovian representation of fractional Brownian motion

We begin with the definition of type I fractional Brownian motion, defined on the whole real line, possessing correlated increments that are in contrast to type II fractional Brownian motion stationary.

**Definition A.1** (Type I Fractional Brownian Motion [23]). *Let $(\Omega, \mathcal{F}, \mathbb{P})$ be a complete probability space equipped with a complete and right continuous filtration $\{\mathcal{F}_t\}$ and $\Gamma$ the Gamma function. For two standard independent $\{\mathcal{F}_t\}$-Brownian motions (BMs) $\tilde{B}$ and $B$ the centered Gaussian process $W^H = (W^H_t)_{t \in \mathbb{R}}$ with*

$$W^H_t := \frac{1}{\Gamma(H + \frac{1}{2})} \int_{-\infty}^{0} ((t-s)^{H-\frac{1}{2}} - (-s)^{H-\frac{1}{2}}) \mathrm{d}\tilde{B}_s + \frac{1}{\Gamma(H + \frac{1}{2})} \int_{0}^{t} (t-s)^{H-\frac{1}{2}} \mathrm{d}B_s \quad (22)$$

*uniquely characterized in law by its covariances*

$$\mathbb{E}\left[W^H_t W^H_s\right] = \frac{1}{2}\left[t^{2H} + s^{2H} - (t-s)^{2H}\right], \quad t \geq s > 0 \quad (23)$$

*is called type I fractional Brownian motion (fBM) with Hurst index $H \in (0,1)$.*

Type II fBM from the main paper is retrieved by setting the additionally defined BM $\tilde{B}$ on the negative real line to zero. Therefore, the difference to type II fBM is the stochastic integral w.r.t. $\tilde{B}$ that yields stationary increments and a non trivial distribution at $t = 0$. For $H = 0.5$, the process is a BM and has thus independent increments. For $H \in (0,1) \setminus \{\frac{1}{2}\}$, the process possesses correlated increments and, compared to BM, smoother paths for $H > 0.5$ due to positively correlated increments (super-diffusion) and rougher paths for $H < 0.5$ due to negatively correlated increments (sub-diffusion). These three regimes reflect for type I fBM in the same change of quadratic variation from $t$ to zero quadratic variation in the smooth regime and to infinite quadratic variation in the rough regime [30]. To prepare the approximation of the non-Markovian and non-semimartingale fBM [25] via Markovian semimartingales, define for every $\gamma \in (0, \infty)$ the Ornstein-Uhlenbeck process $Y^\gamma$ given by

$$Y^\gamma_t := Y^\gamma_0 e^{-t\gamma} + \int_{0}^{t} e^{-\gamma(t-s)} \mathrm{d}B_s, \quad t \geq 0, \quad Y_0 := \int_{-\infty}^{0} e^{s\gamma} \mathrm{d}\tilde{B}_s, \quad (24)$$

with speed of mean reversion $\gamma$ and non trivial starting value in contrast to the OU processes defined in eq. (6) of the main paper. By Itô's product rule [30], the process $Y^\gamma$ solves the same SDE

$$\mathrm{d}Y^\gamma_t = -\gamma Y^\gamma_t \mathrm{d}t + \mathrm{d}B_t, \quad Y_0 = \int_{-\infty}^{0} e^{s\gamma} \mathrm{d}\tilde{B}_s, \quad (25)$$

with different starting value. According to Harms and Stefanovits [26] we represent fBm by an integral over the predefined family of Ornstein-Uhlenbeck processes.

**Theorem A.2** (Markovian representation of fBM [26, 27]). *The non-Markovian process $W^H$ permits the infinite-dimensional Markovian representation*

$$W^H_t = \begin{cases} \int_0^\infty (Y^\gamma_t - Y^\gamma_0)\, \nu_1(\gamma) \mathrm{d}\gamma, & H \leq \frac{1}{2} \\ -\int_0^\infty \partial_\gamma (Y^\gamma_t - Y^\gamma_0)\, \nu_2(\gamma) \mathrm{d}\gamma, & H > \frac{1}{2} \end{cases} \quad (26)$$

*where*

$$\nu_1(\gamma) = \frac{\gamma^{-(H+\frac{1}{2})}}{\Gamma(H + \frac{1}{2})\Gamma(\frac{1}{2} - H)} \quad and \quad \nu_2(\gamma) = \frac{\gamma^{-(H-\frac{1}{2})}}{\Gamma(H + \frac{1}{2})\Gamma(\frac{3}{2} - H)}. \quad (27)$$

Note that we follow Daems et al. [27] in replacing the process $Z^\gamma_t := Z^\gamma_0 e^{-t\gamma} + \int_0^t e^{-(t-s)\gamma} Y^\gamma_s ds$ from the original theorem throughout this work by $Z^\gamma_t = -\partial_\gamma Y^\gamma_t + (\partial_\gamma Y^\gamma_0 + Z^\gamma_0) e^{-t\gamma}$. This is

justified by Harms and Stefanovits [26, Remark 3.5] and simplifies for $H > \frac{1}{2}$ the approximation of fBM and the definition of our generative model, since we only have to reverse the $Y^\gamma$ processes instead of the pairs $(Y^\gamma, Z^\gamma)$. For $Y_0^\gamma = 0$ eq. (26) yields an infinite-dimensional Markovian representation of type II fBM [27]. The MA-fBM from Definition 3.2 in the main paper becomes for type I fBM

$$\hat{B}_t^H = \sum_{k=1}^K \omega_k \left( Y_t^k - Y_0^k \right), \quad H \in (0,1), \quad t \geq 0 \tag{28}$$

with non trivial $\mathbf{Y}_0 = (Y_0^1, ..., Y_0^1)$ that is a centered multivariate Gaussian with covariances $\mathbb{E}\left[Y_0^k Y_0^l\right] = 1/(\gamma_k + \gamma_l)$ [27]. Theorem 3.3 holds true for type I fBM as well with optimal approximation coefficients given in Daems et al. [27, Proposition 5]. For more details on the properties and distinction of type I and type II fBM we refer the reader to Daems et al. [27].

## A.2 The forward model

We define in the following a score-based generative model approximating a fractional diffusion process driven by type I fBM. For the remainder of Appendix A we assume $Y_0^k = \int_{-\infty}^0 e^{s\gamma_k} \mathrm{d}\tilde{B}_s$ for all $1 \leq k \leq K$ where the setting from the main paper with type II fBM is recovered by choosing $Y_0^k = 0$ instead. Let $\hat{\mathbf{B}}^H$ be a $D$-dimensional MA-fBM with Hurst index $H \in (0,1)$. For continuous functions $\mu : [0,T] \to \mathbb{R}$ and $g : [0,T] \to \mathbb{R}$ we define the forward process $\mathbf{X} = (\mathbf{X}_t)_{t \in [0,T]}$ by

$$\mathrm{d}\mathbf{X}_t = \mu(t)\mathbf{X}_t \mathrm{d}t + g(t)\mathrm{d}\hat{\mathbf{B}}_t^H, \quad \mathbf{X}_0 = \mathbf{x}_0 \sim p_0, \quad t \in [0,T] \tag{29}$$

where $p_0$ is an unknown data distribution from which we aim to sample from. Using eq. (25) we note

$$\mathrm{d}\hat{\mathbf{B}}_t^H = -\sum_{k=1}^K \omega_k \gamma_k Y_t^k \mathrm{d}t + \sum_{k=1}^K \omega_k \mathrm{d}\mathbf{B}_t, \tag{30}$$

where $\mathbf{B} = (B_1, ..., B_d)$ is a multivariate BM. With $\bar{\omega} := \sum_{k=1}^K \omega_k$ we rewrite the dynamics of the forward process as

$$\mathrm{d}\mathbf{X}_t = \left[\mu(t)\mathbf{X}_t - g(t)\sum_{k=1}^K \omega_k \gamma_k \mathbf{Y}_t^k\right] \mathrm{d}t + \bar{\omega}g(t)\mathrm{d}\mathbf{B}_t, \quad t \in [0,T], \tag{31}$$

Taking into account the dynamics of the OU processes, we define the *augmented forward process* $\mathbf{Z} = (\mathbf{Z}_t)_{t \in [0,T]}$ by

$$\mathbf{Z}_t = (X_{t,1}, Y_{t,1}^1, ..., Y_{t,1}^K, X_{t,2}, Y_{t,2}^1, ..., Y_{t,2}^K, ..., ..., ..., X_{t,D}, Y_{t,D}^1, ...Y_{t,D}^K) \in \mathbb{R}^{D(K+1)} \tag{32}$$

following the dynamics

$$\mathrm{d}\mathbf{Z}_t = \mathbf{F}(t)\mathbf{Z}_t \mathrm{d}t + \mathbf{G}(t)\mathrm{d}\mathbf{B}_t \tag{33}$$

with $\mathbf{F}(t) = diag(\mathbf{R}(t), ..., \mathbf{R}(t)) \in \mathbb{R}^{D(K+1), D(K+1)}$,

$$\mathbf{R}(t) = \begin{pmatrix} \mu(t) & -g(t)\omega_1\gamma_1 & \cdots & -g(t)\omega_K\gamma_K \\ \mathbf{0}_K & & -diag(\gamma_1, ..., \gamma_K) & \end{pmatrix} \in \mathbb{R}^{K+1, K+1} \tag{34}$$

and

$$\boldsymbol{G}(t) = \left(\bar{\omega}g(t)\boldsymbol{I}_D \quad \boldsymbol{I}_D \quad \cdots \quad \boldsymbol{I}_D\right)^T \in \mathbb{R}^{D(K+1), D}. \tag{35}$$

For each dimension $1 \leq d \leq D$, the dynamics of the process transforming $\mathbf{x}_{0,d}$ reduce to those of the augmented forward process with $D = 1$, given by

$$\mathrm{d}\mathbf{Z}_t = \mathbf{F}(t)\mathbf{Z}_t \mathrm{d}t + \mathbf{G}(t)\mathrm{d}B_t, \tag{36}$$

where the $K+1$ processes that transform $\mathbf{x}_{0,d}$ are all driven by the same one-dimensional BM $B$. The augmented forward process $\mathbf{Z}$ conditioned on $\mathbf{y}_0^1, ..., \mathbf{y}_0^K$ and a data sample $\mathbf{x}_0 \sim p_0$ is a linear transformation of BM and hence a Gaussian process and so is $\mathbf{X}$ [43]. Since the integral w.r.t BM has zero mean, the mean vector of the augmenting processes is $\mathbb{E}\left[\mathbf{Y}_t^k\right] = \mathbf{0}_d$ for all $1 \leq k \leq K$ and the mean of the conditional forward process is the solution of the ODE

$$\partial_t \mathbb{E}\left[\mathbf{X}_t | \mathbf{x}_0\right] = \mu(t)\mathbb{E}\left[\mathbf{X}_t | \mathbf{x}_0\right] \tag{37}$$

and hence the marginal mean

$$\mathbb{E}\left[\mathbf{X}_t|\mathbf{x}_0\right] = c(t)\mathbf{x}_0 \quad with \quad c(t) = \exp\left(\int_0^t \mu(s)ds\right) \tag{38}$$

is not affected by changing the driving noise to MA-fBM. The marginal covariance matrix $\mathbf{\Sigma}_t$ of the conditional augmented forward process can be approximated numerically by solving an ODE, see Appendix B for details. In addition we present a continuous reparameterization of the forward process, resulting for some forward dynamics in a closed form solution of the marginal covariance matrix. Our result generalizes the explicit formula for the perturbation kernel $p_{0t}(\mathbf{x}|\mathbf{x}_0) = \mathcal{N}(\mathbf{x}; c(t)\mathbf{x}_0, c^2(t)\sigma^2(t)\mathbf{I}_d)$ given in Karras et al. [44].

**Proposition A.3** (Continuous Reparameterization Trick). *Let $\mathbf{x}_0$ be a fixed realisation drawn from $p_0$. The forward process $\mathbf{X} = (\mathbf{X}_t)_{t\in[0,T]}$ conditioned on $\mathbf{x}_0$ admits the continuous reparameterization*

$$\mathbf{X}_t = c(t)\left(\mathbf{x}_0 + \int_0^t \alpha(t,s)d\mathbf{B}_s\right) + \underbrace{c(t)\sum_{k=1}^K \omega_k\gamma_k \int_0^t \frac{g(s)}{c(s)}e^{-s\gamma_k}ds\mathbf{Y}_0^k}_{=0 \text{ for type II fBM since } \mathbf{Y}_0^k=0} \tag{39}$$

*with $c(t) = \exp\left(\int_0^t \mu(s)ds\right)$ and*

$$\alpha(t,s) = -\sum_{k=1}^K \omega_k\gamma_k \int_s^t \frac{g(u)}{c(u)}e^{-\gamma_k(u-s)}du + \bar{\omega}\frac{g(s)}{c(s)} \tag{40}$$

*such that $\mathbf{X}_t|\mathbf{x}_0 \sim \mathcal{N}\left(c(t)\mathbf{x}_0, \left[c^2(t)\sigma^2(t) + \sigma_K^2(t)\right]\mathbf{I}_d\right)$ is a Gaussian random vector for all $t \in (0,T]$ with*

$$\sigma^2(t) = \int_0^t \alpha^2(t,s)ds \tag{41}$$

*and*

$$\sigma_K^2 = c^2(t)\sum_{k=1}^K \frac{\gamma_k}{2}\left[\omega_k \int_0^t \frac{g(s)}{c(u)}du\right]^2 \tag{42}$$

$$+ 2c^2(t)\sum_{k<l}\frac{\omega_k\omega_l\gamma_k\gamma_l}{\gamma_k+\gamma_l}\int_0^t \frac{g(s)}{c(s)}e^{-s\gamma_k}ds \int_0^t \frac{g(s)}{c(s)}e^{-s\gamma_l}ds \tag{43}$$

*vanishing for an underlying type II fBM.*

*Proof.* By continuity, the functions $\mu$ and $\sigma$ are bounded. Moreover, the processes $Y_j^1, ..., Y_j^K$ posses continuous, hence bounded, paths and thus

$$\int_0^t |\mu(u)|du < \infty, \quad \int_0^t \sigma^2(u)du < \infty \quad and \quad \int_0^t |\sum_k^K \omega_k\gamma_k\mathbf{Y}_t^k|du < \infty \quad \mathbb{P}-a.s., \tag{44}$$

where the last integral is understood entrywise. Hence, by Cohen and Elliott [30, Theorem 16.6.1], the unique solution of the SDE eq. (31) is given explicitly as

$$\mathbf{X}_t = c(t)\left(\mathbf{x}_0 - \int_0^t \frac{g(u)}{c(u)}\left[\sum_{k=1}^K \omega_k\gamma_k\mathbf{Y}_u^k\right]du + \bar{\omega}\int_0^t \frac{g(u)}{c(u)}d\mathbf{B}_u\right), \tag{45}$$

with $c(t) = \exp\left(\int_0^t \mu(s)ds\right)$. Define

$$\mathbf{J}\left(\mathbf{Y}_0^{[K]}, t\right) := \sum_{k=1}^K \omega_k\gamma_k \int_0^t \frac{g(s)}{c(s)}e^{-s\gamma_k}ds\mathbf{Y}_0^k \tag{46}$$

and by the definition of $Y_j^k$ in (24) we calculate using the Stochastic Fubini Theorem [26]

$$\int_0^t \frac{g(u)}{c(u)} \left[ \sum_{k=1}^K \omega_k \gamma_k \mathbf{Y}_u^k \right] \mathrm{d}u = \sum_{k=1}^K \omega_k \gamma_k \int_0^t \int_0^u \frac{g(u)}{c(u)} e^{-\gamma_k(u-s)} \mathrm{d}\mathbf{B}_s \mathrm{d}u + \mathbf{J}(\mathbf{Y}_0^{[K]}, t) \qquad (47)$$

$$= \int_0^t \sum_{k=1}^K \omega_k \gamma_k \int_s^t \frac{g(u)}{c(u)} e^{-\gamma_k(u-s)} \mathrm{d}u \mathrm{d}\mathbf{B}_s + \mathbf{J}\left(\mathbf{Y}_0^{[K]}, t\right) \qquad (48)$$

and hence

$$\mathbf{X}_t = c(t) \left( \mathbf{x}_0 - \int_0^t \frac{g(u)}{c(u)} \left[ \sum_{k=1}^K \omega_k \gamma_k \mathbf{Y}_u^k \right] \mathrm{d}u + \bar{\boldsymbol{\omega}} \int_0^t \frac{g(u)}{c(u)} \mathrm{d}\mathbf{B}_u \right) \qquad (49)$$

$$= c(t) \left( \mathbf{x}_0 - \int_0^t \sum_{k=1}^K \omega_k \gamma_k \int_s^t \frac{g(u)}{c(u)} e^{-\gamma_k(u-s)} \mathrm{d}u \mathrm{d}\mathbf{B}_s + \bar{\boldsymbol{\omega}} \int_0^t \frac{g(u)}{c(u)} \mathrm{d}\mathbf{B}_u - \mathbf{J}\left(\mathbf{Y}_0^{[K]}, t\right) \right)$$

$$= c(t) \left( \mathbf{x}_0 + \int_0^t \left[ -\sum_{k=1}^K \omega_k \gamma_k \int_s^t \frac{g(u)}{c(u)} e^{-\gamma_k(u-s)} \mathrm{d}u + \bar{\boldsymbol{\omega}} \frac{g(s)}{c(s)} \right] \mathrm{d}\mathbf{B}_s - \mathbf{J}\left(\mathbf{Y}_0^{[K]}, t\right) \right)$$

$$= c(t)\mathbf{x}_0 + c(t) \int_0^t \int_0^t \alpha(t,s) \mathrm{d}\mathbf{B}_s - c(t)\mathbf{J}\left(\mathbf{Y}_0^{[K]}, t\right) \qquad (50)$$

with

$$\alpha(t,s) = -\sum_{k=1}^K \omega_k \gamma_k \int_s^t \frac{g(u)}{c(u)} e^{-\gamma_k(u-s)} \mathrm{d}u + \bar{\boldsymbol{\omega}} \frac{g(s)}{c(s)}. \qquad (51)$$

Since $\alpha(t, \cdot)$ is continuous for every fixed $t \in [0, T]$ we have $\int_0^t \alpha^2(t,s)\mathrm{d}s < \infty$. Using that the integral of a bounded deterministic function w.r.t. Brownian motion is a Gaussian process we have by Itô's isometry

$$\int_0^t \alpha(t,s) \mathrm{d}\mathbf{B}_s \sim \mathcal{N}\left(\mathbf{0}_d, \sigma^2(t)\mathbf{I}_d\right) \quad with \quad \sigma^2(t) = \int_0^t \alpha^2(t,s)\mathrm{d}s. \qquad (52)$$

Therefore, conditional on $\mathbf{x}_0$, the random vector $\mathbf{X}_t$ is Gaussian with mean vector

$$\boldsymbol{m}_t^{\mathbf{x}} = c(t)\boldsymbol{x}_0 + \underbrace{\mathbb{E}\left[\mathbf{J}(\mathbf{Y}_0^{[K]})\right]}_{=0} = \mathbf{x}_0 \exp\left(\int_0^t \mu(s)\mathrm{d}s\right). \qquad (53)$$

Moreover, $\tilde{B}_j$ and $B_j$ corresponding to the entries of $\tilde{\mathbf{B}} = (\tilde{B}_1, ..., \tilde{B}_d)$ and $\mathbf{B} = (B_1, ..., B_d)$ are independent by Theorem A.1 resulting in the entrywise variance

$$\Sigma_{t,j,j}^{\mathbf{x}} = c^2(t) \int_0^t \alpha^2(t,s)\mathrm{d}s + \sigma_K^2(t) \qquad (54)$$

with

$$\sigma_K^2(t) = \mathbb{V}\left[\mathbf{J}(\mathbf{Y}_0^{[K]})_j\right] = c^2(t) \sum_{k=1}^K \frac{\gamma_k}{2} \left[\omega_k \int_0^t \frac{g(s)}{c(u)}\mathrm{d}u\right]^2 \qquad (55)$$

$$+ 2c^2(t) \sum_{k<l} \frac{\omega_k \omega_l \gamma_k \gamma_l}{\gamma_k + \gamma_l} \int_0^t \frac{g(s)}{c(s)} e^{-s\gamma_k}\mathrm{d}s \int_0^t \frac{g(s)}{c(s)} e^{-s\gamma_l}\mathrm{d}s, \qquad (56)$$

where we used again Itô's isometry to calculate

$$\mathbb{E}\left[Y_{0,j}^k Y_{0,j}^l\right] = \mathbb{E}\left[\int_{-\infty}^0 e^{\gamma_k s}\mathrm{d}\tilde{B}_{s,j} \int_{-\infty}^0 e^{\gamma_l s}\mathrm{d}\tilde{B}_{s,j}\right] = \int_{-\infty}^0 e^{(\gamma_k + \gamma_l)s}\mathrm{d}s = \frac{1}{\gamma_k + \gamma_l}. \qquad (57)$$

Since the entries of $\mathbf{B}$ are independent, we find the covariance matrix

$$\Sigma_t^{\mathbf{x}} = \left[c^2(t)\sigma^2(t) + \sigma_K^2(t)\right] \mathbf{I}_d. \qquad (58)$$

$\square$

The preceding proposition generalizes the "reparameterization trick"[3] from discrete time to continuous-time in the sense that

$$\mathbf{X}_{t_n} = \sqrt{\bar{\alpha}_{t_n}}\mathbf{x}_0 + \sqrt{1 - \bar{\alpha}_{t_n}}\boldsymbol{\epsilon}, \quad \boldsymbol{\epsilon} \sim \mathcal{N}(\mathbf{0}_d, \mathbf{I}_d) \tag{59}$$

used in discrete time [2] with time steps $0 = t_0 < ... < t_N = T$ is replaced by our continuous-time reparameterization

$$\mathbf{X}_t = c(t)\left(\mathbf{x}_0 + \int_0^t \alpha(t,s)\mathrm{d}\mathbf{B}_s\right) + c(t)\sum_{k=1}^K \omega_k \gamma_k \int_0^t \frac{g(s)}{c(s)}e^{-s\gamma_k}\mathrm{d}s\mathbf{Y}_0^k, \tag{60}$$

enabling to directly sample $\mathbf{X}_t|\mathbf{x}_0 \sim \mathcal{N}(c(t)\mathbf{x}_0 + \left[c^2(t)\sigma^2(t) + \sigma_K^2(t)\right]\mathbf{I}_D)$ for a given data sample $\mathbf{x}_0$ and time point $t \in (0,T]$, in case that $\sigma^2(t)$ and $\sigma_K^2(t)$ have a closed form solution. For a complete characterization of the marginal covariance matrix $\mathbf{\Sigma}_t$ of the conditioned augmented forward process we calculate by Itô isometry with $X = X_j$ and $Y^l = Y_j^l$ for all $1 \le j \le D$, $1 \le l \le K$ and any $t \in [0,T]$

$$\mathbb{E}\left[X_tY_t^l\right] = c(t)\int_0^t \alpha(t,s)e^{-\gamma_k(t-s)}\mathrm{d}s + c(t)\sum_{l=1}^K \frac{\omega_k\gamma_k}{\gamma_k + \gamma_l}e^{-\gamma_l t}\int_0^t \frac{g(s)}{c(s)}e^{-s\gamma_k}\mathrm{d}s \tag{61}$$

and

$$\mathbb{E}\left[Y_t^kY_t^l\right] = \frac{e^{-(\gamma_k+\gamma_l)s}}{\gamma_k + \gamma_l} + \frac{1 - e^{-(\gamma_k+\gamma_l)t}}{\gamma_k + \gamma_l} = \frac{1}{\gamma_k + \gamma_l} \tag{62}$$

reducing for type II fBM to

$$\mathbb{E}\left[X_tY_t^l\right] = c(t)\int_0^t \alpha(t,s)e^{-\gamma_k(t-s)}\mathrm{d}s \quad and \quad \mathbb{E}\left[Y_t^kY_t^l\right] = \frac{1 - e^{-(\gamma_k+\gamma_l)t}}{\gamma_k + \gamma_l}. \tag{63}$$

We denote in the following the stacked vector of the augmenting processes by

$$\mathbf{Y}_t^{[K]} = (Y_{t,1}^1, Y_{t,1}^2, ..., Y_{t,1}^K, Y_{t,2}^1, Y_{t,2}^2, ..., Y_{t,2}^K, ...., Y_{t,D}^1, Y_{t,D}^2, ..., Y_{t,D}^K) \in \mathbb{R}^{D(K+1)}. \tag{64}$$

The random vector $\mathbf{Y}_t^{[K]}$ is a centered Gaussian process with covariance matrix

$$\mathbf{\Lambda}_t = diag(\mathbf{\Sigma}_t^{\mathbf{y}}, ..., \mathbf{\Sigma}_t^{\mathbf{y}}) \in \mathbb{R}^{D\cdot K, D\cdot K}, \quad \mathbf{\Sigma}_t^{\mathbf{y}} \in \mathbb{R}^{K,K}, \quad [\mathbf{\Sigma}_t^{\mathbf{y}}]_{k,l} = \mathbb{E}\left[Y_t^kY_t^l\right] \tag{65}$$

where $\mathbf{\Sigma}_t^{\mathbf{y}}$ does not depend on the dimension $1 \le j \le D$ and we write $q_t$ for the multivariate Gaussian density of $\mathbf{Y}_t^{[K]}$. Since we know the distribution of $\mathbf{Y}_0^{[K]}$, we can directly calculate the corresponding score function by

$$\nabla_{\mathbf{y}^{[K]}}\log q_t\left(\mathbf{Y}_t^{[K]}\right) = -\mathbf{\Lambda}_t^{-1}\mathbf{Y}_t^{[K]}. \tag{66}$$

## A.3 Estimating the score via augmented score matching loss

Conditioning $\mathbf{Z}_t$ on $\mathbf{x}_0 \sim p_0$ and a realisation $\mathbf{y}_t^{[K]}$ of the stacked augmenting processes $\mathbf{Y}_t^{[K]}$ defined in eq. (64) at fixed time $t \in [0,T]$ results in the Gaussian vector $\tilde{\mathbf{X}}_t \sim \mathcal{N}(\tilde{\boldsymbol{m}}_t, \tilde{\mathbf{\Sigma}}_t)$ with mean

$$\tilde{\boldsymbol{m}}_t = c(t)\mathbf{x}_0 + \sum_{k=1}^K \eta_t^k\mathbf{y}_t^k, \quad where \quad \eta_t^k = \sum_{l=1}^K \mathbb{E}\left[X_tY_t^l\right]\left[(\mathbf{\Sigma}_t^{\mathbf{y}})^{-1}\right]_{l,k} \tag{67}$$

and covariance

$$\tilde{\mathbf{\Sigma}}_t = \left(c^2(t)\sigma^2(t) - \tau_t^2\right)\mathbf{I}_d, \quad where \quad \tau_t^2 = \sum_{k=1}^K \eta_t^k\mathbb{E}\left[X_tY_t^k\right]. \tag{68}$$

We denote with $\nabla_{\mathbf{x}}\log p_{0t}$ the conditional score function of $\tilde{\mathbf{X}}_t$ and calculate for the gradient w.r.t. $\mathbf{x} = (x_1, ..., x_D) \in \mathbb{R}^D$

$$\nabla_{\mathbf{x}}\log p_{0t}(\mathbf{x}|\mathbf{y}_t^{[K]}, \mathbf{x}_0) = -\tilde{\mathbf{\Sigma}}_t^{-1}(\mathbf{x} - \tilde{\boldsymbol{m}}_t) = -\frac{(\mathbf{x} - \tilde{\boldsymbol{m}}_t)}{(c^2(t)\sigma^2(t) - \tau_t^2)}. \tag{69}$$

---

[3]See https://lilianweng.github.io/posts/2021-07-11-diffusion-models/ for the derivation in discrete time.

and for the gradient w.r.t. $\mathbf{y}^k = (y_1^k, ..., y_D^k) \in \mathbb{R}^D$

$$\nabla_{\mathbf{y}^k} \log p_{0t}(\mathbf{x}|\mathbf{y}_t^{[K]}, \mathbf{x}_0) = -\frac{1}{2} \nabla_{\mathbf{y}^k} \left[ (\mathbf{x} - \tilde{\boldsymbol{m}}_t)^T \tilde{\boldsymbol{\Sigma}}_t^{-1} (\mathbf{x} - \tilde{\boldsymbol{m}}_t) \right] \tag{70}$$

$$= -\eta_t^k \nabla_{\mathbf{x}} \log p_{0t}(\mathbf{x}|\mathbf{y}_t^{[K]}, \mathbf{x}_0). \tag{71}$$

Deploying this relation of $\nabla_{\mathbf{x}} \log p_{0t}$ and $\nabla_{\mathbf{y}^k} \log p_{0t}$ we derive the *augmenting score matching loss* that reduces the dimensionality of the score model we have to learn to the dimensionality of the data distribution and results in a score model guided by the the known score function $\nabla_{\mathbf{y}^{[K]}} \log q_t$.

**Proposition A.4** (Optimal Score Model). *Assume that $s_{\boldsymbol{\theta}}$ is optimal w.r.t. the augmented score matching loss $\mathcal{L}$. The score model*

$$S_{\boldsymbol{\theta}}(\mathbf{Z}_t, t) := \left( s_{\boldsymbol{\theta}}(\mathbf{X}_t - \sum_k \eta_t^k \mathbf{Y}_t^k, t), -\eta_t^1 s_{\boldsymbol{\theta}}(\mathbf{X}_t - \sum_k \eta_t^k \mathbf{Y}_t^k, t), ..., -\eta_t^K s_{\boldsymbol{\theta}}(\mathbf{X}_t - \sum_k \eta \mathbf{Y}_t^k, t) \right)$$

*yields the optimal $L^2(\mathbb{P})$ approximation of $\nabla_{\mathbf{z}} \log p_t(\mathbf{Z}_t)$ via*

$$S_{\boldsymbol{\theta}}(\mathbf{Z}_t, t) + \nabla_{\mathbf{z}} \log q_t(\mathbf{Y}_t^{[K]}) \approx \nabla_{\mathbf{z}} \log p_t(\mathbf{Z}_t). \tag{72}$$

*Proof.* Fix $t \in [0, T]$. We write $p_t^{aug}$ for the density of $\mathbf{Z}_t$, $p_{0t}^{aug}$ for the conditional density of $\mathbf{Z}_t$ on $\mathbf{X}_0$, $p_{0t}$ for the density of $\tilde{\mathbf{X}}_t$ and $q_{0t}$ for the conditional density of $\mathbf{Y}_t^{[K]}$ on $\mathbf{X}_0$. First note that $\mathbf{Y}_t^{[K]}$ and $\mathbf{X}_0$ are independent by assumption and hence $q_t = q_{0t}$. By direct calculations we find

$$\nabla_{\mathbf{x}} \log p_t^{aug}(\mathbf{Z}_t) = \mathbb{E}_{(\mathbf{X}_0|\mathbf{X}_t, \mathbf{Y}_t^{[K]})} \left[ \nabla_{\mathbf{x}} \log p_{0t}^{aug}(\mathbf{Z}_t|\mathbf{X}_0) \right] \tag{73}$$

$$= \mathbb{E}_{(\mathbf{X}_0|\mathbf{X}_t, \mathbf{Y}_t^{[K]})} \left[ \nabla_{\mathbf{x}} \log \left( p_{0t}(\mathbf{X}_t|\mathbf{Y}_t^{[K]}, \mathbf{X}_0) q_{0t}(\mathbf{Y}_t^{[K]}|\mathbf{X}_0) \right) \right] \tag{74}$$

$$= \mathbb{E}_{(\mathbf{X}_0|\mathbf{X}_t, \mathbf{Y}_t^{[K]})} \left[ \nabla_{\mathbf{x}} \log p_{0t}(\mathbf{X}_t|\mathbf{Y}_t^{[K]}, \mathbf{X}_0) + \underbrace{\nabla_{\mathbf{x}} \log q_t(\mathbf{Y}_t^{[K]})}_{=\mathbf{0}_d} \right] \tag{75}$$

$$= \mathbb{E}_{(\mathbf{X}_0|\mathbf{X}_t, \mathbf{Y}_t^{[K]})} \left[ \nabla_{\mathbf{x}} \log p_{0t}(\mathbf{X}_t|\mathbf{Y}_t^{[K]}, \mathbf{X}_0) \right] \tag{76}$$

$$\overset{(69)}{=} \mathbb{E}_{(\mathbf{X}_0|\mathbf{X}_t, \mathbf{Y}_t^{[K]})} \left[ \frac{\mathbf{X}_t - \sum_k \eta_t^k \mathbf{Y}_t^k - c(t)\mathbf{X}_0}{c^2(t)\sigma^2(t) - \tau_t^2} \right]. \tag{77}$$

Hence the best $L^2(\mathbb{P})$-approximation of $\nabla_{\mathbf{x}} \log p_t^{aug}(\mathbf{Z}_t)$ is a minimizer of the augmented score matching loss by

$$\nabla_{\mathbf{x}} \log p_t^{aug}(\mathbf{Z}_t) \overset{(77)}{=} \mathbb{E}_{(\mathbf{X}_0|\mathbf{X}_t, \mathbf{Y}_t^{[K]})} \left[ \frac{\mathbf{X}_t - \sum_k \eta_t^k \mathbf{Y}_t^k - c(t)\mathbf{X}_0}{c^2(t)\sigma^2(t) - \tau_t^2} \right] \tag{78}$$

$$= \underset{s_{\boldsymbol{\theta}}}{\arg\min} \, \mathbb{E}_{(\mathbf{X}_0, \mathbf{Y}_t^{[K]})} \mathbb{E}_{(\mathbf{X}_t|\mathbf{Y}_t^{[K]}, \mathbf{X}_0)} \left[ \left\| s_{\boldsymbol{\theta}}(\mathbf{X}_t - \sum_{k=1}^K \eta_t^k \mathbf{Y}_t^k, t) - \frac{\mathbf{X}_t - \sum_k \eta_t^k \mathbf{Y}_t^k - c(t)\mathbf{X}_0}{c^2(t)\sigma^2(t) - \tau_t^2} \right\|^2 \right] \tag{79}$$

$$\overset{(69)}{=} \underset{s_{\boldsymbol{\theta}}}{\arg\min} \, \mathbb{E}_{(\mathbf{X}_0, \mathbf{Y}_t^{[K]})} \mathbb{E}_{(\mathbf{X}_t|\mathbf{Y}_t^{[K]}, \mathbf{X}_0)} \left[ \left\| s_{\boldsymbol{\theta}}(\mathbf{X}_t - \sum_{k=1}^K \eta_t^k \mathbf{Y}_t^k, t) - \nabla_{\mathbf{x}} \log p_{0t}(\mathbf{X}_t|\mathbf{Y}_t^{[K]}, \mathbf{X}_0) \right\|^2 \right] \tag{80}$$

Assume now that $s_{\boldsymbol{\theta}}$ is a minimizer of the augmented score matching loss. Similar to the calculation above we have

$$\nabla_{\mathbf{y}^k} \log p_t^{aug}(\mathbf{Z}_t) = \mathbb{E}_{(\mathbf{X}_0|\mathbf{X}_t, \mathbf{Y}_t^{[K]})} \left[ \nabla_{\mathbf{y}^k} \log p_{0t}^{aug}(\mathbf{Z}_t|\mathbf{X}_0) \right] \tag{81}$$

$$= \mathbb{E}_{(\mathbf{X}_0|\mathbf{X}_t, \mathbf{Y}_t^{[K]})} \left[ \nabla_{\mathbf{y}^k} \log \left( p_{0t}(\mathbf{X}_t|\mathbf{Y}_t^{[K]}, \mathbf{X}_0) q_{0t}(\mathbf{Y}_t^{[K]}|\mathbf{X}_0) \right) \right] \tag{82}$$

$$= \mathbb{E}_{(\mathbf{X}_0|\mathbf{X}_t, \mathbf{Y}_t^{[K]})} \left[ \nabla_{\mathbf{y}^k} \log p_{0t}(\mathbf{X}_t|\mathbf{Y}_t^{[K]}, \mathbf{X}_0) + \nabla_{\mathbf{y}^k} \log q_t(\mathbf{Y}_t^{[K]}) \right] \tag{83}$$

$$\overset{(70)}{=} -\eta_t^k \mathbb{E}_{(\mathbf{X}_0|\mathbf{X}_t, \mathbf{Y}_t^{[K]})} \left[ \nabla_{\mathbf{x}} \log p_{0t}(\mathbf{X}_t|\mathbf{Y}_t^{[K]}, \mathbf{X}_0) \right] + \nabla_{\mathbf{y}^k} \log q_t(\mathbf{Y}_t^{[K]}) \tag{84}$$

and hence $-\eta_t^k s_{\boldsymbol{\theta}}(\mathbf{X}_t - \sum_k \eta_t^k \mathbf{Y}_t^k) + \nabla_{\mathbf{y}^k} \log q_t(\mathbf{Y}_t^{[K]})$ is the best approximation of $\nabla_{\mathbf{y}^k} \log p_t^{aug}(\mathbf{Z}_t)$ in $L^2(\mathbb{P})$ and the score model

$$S_{\boldsymbol{\theta}}(\mathbf{Z}_t, t) := \left( s_{\boldsymbol{\theta}}(\mathbf{X}_t - \sum_k \eta_t^k \mathbf{Y}_t^k, t), -\eta_t^1 s_{\boldsymbol{\theta}}(\mathbf{X}_t - \sum_k \eta_t^k \mathbf{Y}_t^k, t), ..., -\eta_t^K s_{\boldsymbol{\theta}}(\mathbf{X}_t - \sum_k \eta \mathbf{Y}_t^k, t) \right)$$

yields the best $L^2(\mathbb{P})$-approximator of $\nabla_{\mathbf{z}} \log p_t$ via

$$S_{\boldsymbol{\theta}}(\mathbf{Z}_t, t) + \nabla_{\mathbf{z}} \log q_t(\mathbf{Y}_t^{[K]}) \approx \nabla_{\mathbf{z}} \log p_t(\mathbf{Z}_t). \tag{85}$$

$\square$

## B  Forward sampling

We assume throughout this section type II fBM. Given the marginal covariance matrix $\boldsymbol{\Sigma}_t$ of $\mathbf{Z}_t | \mathbf{x}_0$ we uniformly sample first a time point $t \in (0, T]$ and second $\mathbf{Z}_t \sim \mathcal{N}(\hat{\mathbf{z}}_t, \boldsymbol{\Sigma}_t)$ with

$$\hat{\mathbf{z}}_t = (c(t)\mathbf{x}_{0,1}, 0, ..., 0, c(t)\mathbf{x}_{0,2}, 0, ..., 0, ..., ..., ..., c(t)\mathbf{x}_{0,D}, 0, ...0) \in \mathbb{R}^{D(K+1)} \tag{86}$$

where we use $\mathbb{E}[\mathbf{X}_t | \mathbf{x}_0] = c(t)\mathbf{x}_0$ and $\mathbb{E}[\mathbf{Y}_t^k] = \mathbf{0}_D$. In the following we characterize further the entries of the marginal covariance matrix $\boldsymbol{\Sigma}_t$. The calculations in this section are straightforward; nevertheless, we present them in full detail to facilitate easy understanding for the interested reader. We begin with rewriting $\sigma^2$ from Proposition 4.2 given by

$$\sigma^2(t) = c^2(t) \int_0^t \alpha^2(t, s) ds \tag{87}$$

where

$$\alpha(t, s) = \bar{\omega} \frac{g(s)}{c(s)} - \sum_{k=1}^{K} \omega_k \gamma_k \int_s^t \frac{g(u)}{c(u)} e^{-\gamma_k(u-s)} du \tag{88}$$

$$= \sum_{k=1}^{K} \omega_k \underbrace{\left( \frac{g(s)}{c(s)} - \gamma_k \int_s^t \frac{g(u)}{c(u)} e^{-\gamma_k(u-s)} du \right)}_{=: \alpha_k(t,s)}. \tag{89}$$

With

$$f_k(u, s) := \frac{g(u)}{c(u)} e^{-\gamma_k(u-s)} \quad and \quad I_k(t, s) := \int_s^t f_k(u, s) du \tag{90}$$

we have

$$\sigma_t^2 = c^2(t) \int_0^t \alpha^2(t, s) ds \tag{91}$$

$$= c^2(t) \int_0^t \left[ \sum_{k=1}^{K} \omega_k \left( \frac{g(s)}{c(s)} - \gamma_k \int_s^t f_k(u, s) du \right) \right]^2 ds \tag{92}$$

$$= c^2(t) \int_0^t \left( \sum_{k=1}^{K} \omega_k \alpha_k(t, s) \right)^2 ds \tag{93}$$

$$= c^2(t) \int_0^t \sum_{i=1,j=1}^{K} \omega_i \omega_j \alpha_i(t, s) \alpha_j(t, s) ds \tag{94}$$

$$= \sum_{i=1,j=1}^{K} \omega_i \omega_j c^2(t) \int_0^t \alpha_i(t, s) \alpha_j(t, s) ds \tag{95}$$

$$= \sum_{i=1,j=1}^{K} \omega_i \omega_j c^2(t) \int_0^t \left( \frac{g(s)}{c(s)} - \gamma_i I_i(t, s) \right) \left( \frac{g(s)}{c(s)} - \gamma_j I_j(t, s) \right) ds \tag{96}$$

$$= \sum_{i,j=1}^{K} \omega_i \omega_j \left\{ \text{var}_B(t) - c^2(t) \int_0^t \left[ \frac{g(s)}{c(s)} (\gamma_i I_i(t, s) + \gamma_j I_j(t, s)) - \gamma_i \gamma_j I_i(t, s) I_j(t, s) \right] ds \right\}, \tag{97}$$

where

$$\text{var}_B(t) = c^2(t) \int_0^t \frac{g^2(s)}{c^2(s)} ds \tag{98}$$

corresponds to the purely Brownian marginal variance, explicitly calculated for VE and VP in Song et al. [16]. Using the above derivation, we derive the closed form variance schedule for FVE dynamics.

**Fractional Variance Exploding** Fix $\sigma_{\max} > \sigma_{\min} > 0$ and define $r := \frac{\sigma_{\max}}{\sigma_{\min}}$. Following Song et al. [16] we set

$$\mu(t) \equiv 0 \quad and \quad g(t) = ar^t \quad with \quad a = \sigma_{min}\sqrt{2\log(r)} \tag{99}$$

such that $c(t) = \exp(0) = 1$ and calculate

$$I_k(t,s) = \int_s^t f_k(u,s)du = \int_s^t ar^u e^{-\gamma_k(u-s)}du = F(t) - F(s) \tag{100}$$

$$= \underbrace{\frac{a}{\ln(r) - \gamma_k}}_{a_k} \left( e^{\ln(r)t - \gamma_k t + \gamma_k s} - e^{\ln(r)s} \right) = a_k \left( r^t e^{-\gamma_k(t-s)} - r^s \right), \tag{101}$$

since the derivative of $F(u) = a_k r^u e^{-\gamma_k(u-s)}$ is given by

$$\frac{d}{du} F(u) = \frac{d}{du}\left[ a_k r^u e^{-\gamma_k(u-s)} \right] = a_k r^u \ln(r) e^{-\gamma_k(u-s)} + a_k r^u e^{-\gamma_k(u-s)}(-\gamma_k) \tag{102}$$

$$= \frac{a}{\ln(r) - \gamma_k}(\ln(r) - \gamma_k)(r^u e^{-\gamma_k(u-s)}) = ar^u e^{-\gamma_k(u-s)}. \tag{103}$$

We calculate for the variance of $X_t|\mathbf{x}_0$

$$\mathbb{V}[X_t|\mathbf{x}_0] = \sum_{i,j=1}^K \omega_i \omega_j \{ \text{var}_B(t) - a\gamma_i \underbrace{\int_0^t r^s I_i(t,s)ds}_{J_i(t)} - a\gamma_j \underbrace{\int_0^t r^s I_j(t,s)ds}_{J_j(t)} \tag{104}$$

$$+ \gamma_i \gamma_j \underbrace{\int_0^t I_i(t,s)I_j(t,s)ds}_{=J_{i,j}(t)} \} \tag{105}$$

with

$$J_k(t) = a_k \int_0^t r^s \left( r^t e^{-\gamma_k(t-s)} - r^s \right) ds = a_k \int_0^t r^{t+s} e^{-\gamma_k(t-s)} ds - a_k \int_0^t r^{2s} ds \tag{106}$$

$$= a_k [F_1(t) - F_1(0)] - a_k [F_2(t) - F_2(0)] = a_k \left[ \frac{r^{2t} - r^t e^{-\gamma_k t}}{\ln(r) + \gamma_k} - \frac{r^{2t} - 1}{2\ln(r)} \right], \tag{107}$$

since

$$\frac{d}{ds} F_1(s) = \frac{\left( r^{t+s} \ln(r) e^{-\gamma_k(t-s)} + r^{t+s} e^{-\gamma_k(t-s)}(\gamma_k) \right)}{\ln(r) + \gamma_k} = r^{t+s} e^{-\gamma_k(t-s)}, \tag{108}$$

$$\frac{d}{ds} F_2(s) = \frac{d}{ds}\left[ \frac{r^{2s}}{2\ln(r)} \right] = \frac{r^{2s}\ln(r)2}{2\ln(r)} = r^{2s}. \tag{109}$$

Finally

$$J_{i,j}(t) = a_i a_j \int_0^t \left( r^t e^{-\gamma_i(t-s)} - r^s \right) \left( r^t e^{-\gamma_j(t-s)} - r^s \right) ds \tag{110}$$

$$= a_i a_j \left[ \left( \frac{r^{2t}\left(1 - e^{-t(\gamma_i + \gamma_j)}\right)}{\gamma_i + \gamma_j} \right) - \frac{r^{2t} - r^t e^{-\gamma_i t}}{\gamma_i + \ln(r)} - \frac{r^{2t} - r^t e^{-\gamma_j t}}{\gamma_j + \ln(r)} + \frac{r^{2t} - 1}{2\ln(r)} \right]. \tag{111}$$

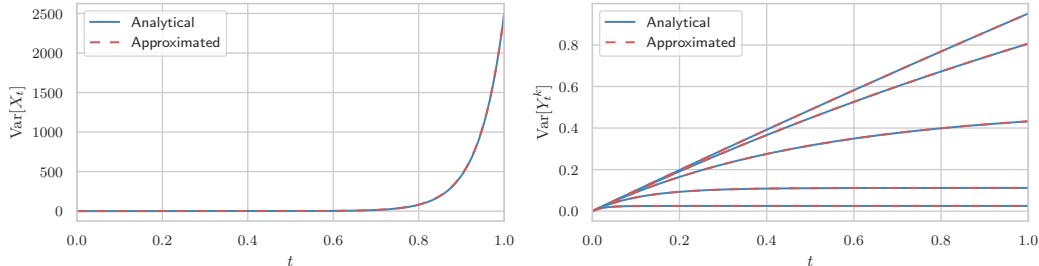

(a) Variance schedule of the forward FVE process.  (b) Variance schedule of the augmenting processes.

Figure 5: Analytical solution (blue) used by our method for FVE dynamics with $K = 5$ and $H = 0.5$ compared to the approximated solution (dashed red) resulting from solving ODE (119).

We calculate the covariance of $X_t|\mathbf{x}_0$ and $Y_t^l$

$$cov(X_t|\mathbf{x}_0, Y_t^l) = c(t) \int_0^t \alpha(t, s) e^{-\gamma_l(t-s)} ds \tag{112}$$

$$= \int_0^t \sum_{k=1}^K \omega_k \left[ \frac{g(s)}{c(s)} - \gamma_k \int_s^t f_k(u, s) du \right] e^{-\gamma_l(t-s)} ds \tag{113}$$

$$= \sum_{k=1}^K \omega_k \left[ a \int_0^t r^s e^{-\gamma_l(t-s)} ds - \gamma_k \int_0^t \int_s^t f_k(u, s) du\, e^{-\gamma_l(t-s)} ds \right] \tag{114}$$

$$= \sum_{k=1}^K \omega_k \left[ a \int_0^t r^s e^{-\gamma_l(t-s)} ds - \gamma_k a_k \int_0^t \left( r^t e^{-\gamma_k(t-s)} - r^s \right) e^{-\gamma_l(t-s)} ds \right] \tag{115}$$

$$= \sum_{k=1}^K \omega_k \left[ a e^{-\gamma_l t} \int_0^t r^s e^{\gamma_l s} ds - \gamma_k a_k \int_0^t \left( r^t e^{-\gamma_k(t-s)} - r^s \right) e^{-\gamma_l(t-s)} ds \right] \tag{116}$$

$$= \sum_{k=1}^K \omega_k \left[ (a + a_k \gamma_k) \frac{(r^t - e^{-\gamma_l t})}{\gamma_l + \ln(r)} - \gamma_k a_k \frac{r^t \left( 1 - e^{-t(\gamma_k + \gamma_l)} \right)}{\gamma_k + \gamma_l} \right]. \tag{117}$$

**Fractional Variance Preserving** To the best of our knowledge, there is no closed form solution for $\int_s^t f_k(u, s) du$ for the dynamics of FVP. In this case, we numerically solve an ODE to determine the marginal covariance matrix of the conditional augmented forward process.

**General Dynamics**. The covariance matrix of the conditional augmented forward process with dynamics

$$d\mathbf{Z}_t = \mathbf{F}(t)\mathbf{Z}_t dt + \mathbf{G}(t)d\mathbf{B}_t, \tag{118}$$

solves the ODE

$$\partial_t \boldsymbol{\Sigma}_t = \mathbf{F}(t)\boldsymbol{\Sigma}_t + \boldsymbol{\Sigma}_t \mathbf{F}(t)^T + \mathbf{G}(t)\mathbf{G}(t)^T, \tag{119}$$

lacking in general a closed form solution [43] in contrast to the setting of Song et al. [16]. This approach is applicable for any choice of $\mu$ and $g$ in the forward dynamics, but depending on the choice of drift and diffusion function it might not yield a numerically stable solution. We empirically observe in Figure 5 that the analytical solution for FVE and the numerical approximation of the variance schedule, determined by solving eq. (119) do not differ significantly.

**Variance schedules**. We normalize the variance schedule of FVE and FVP dynamics such that the variance at $t = 0$ and at $t = T$ is equal to the variance used in the purely Brownian setting of VE and VP dynamics. For both FVE and FVP dynamics we calculate $\tilde{\omega}$ according to Proposition 3.3 and determine $\tilde{\sigma}_T^2$ and define $\boldsymbol{\omega} = \tilde{\omega}/\tilde{\sigma}_T^2$ to weight the OU-processes. By doing so, the terminal variance remains the same throughout different choices of $H$, as empirically confirmed in Figure 6.

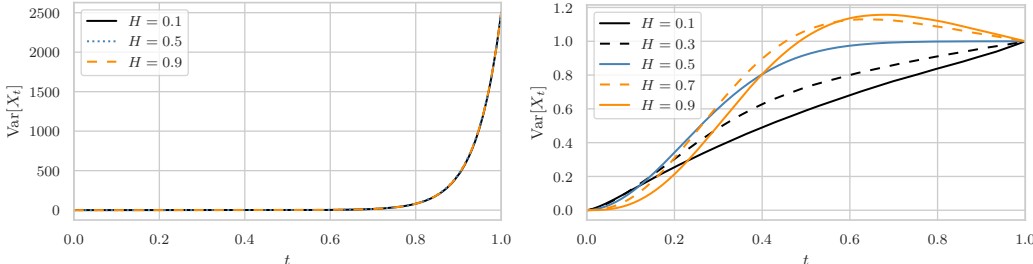

(a) Variance schedules of the forward FVE process.    (b) Variance schedules of the forward FVP process.

Figure 6: Normalized variance schedules for $K = 5$ over time. (a) Variance schedules of FVE dynamics, calculated in closed form according to the derived formulas. The shape of the schedule is preserved throughout different values of $H$. (b) Variance schedules of FVP dynamics numerically approximated. The shape of the schedule is shifted for different values of $H$.

In Figure 6 we observe for FVE dynamics that not only the terminal variance is the same across different choices of $H$ but also the shape of the variance schedule. For FVP dynamics, the shape of the variance schedule shifts with different values of $H$, approaching a nearly linear schedule for $H = 0.1$, while $H = 0.9$ offers a decreasing variance towards the end near $t = T$.

## C    Implementation details

We used for all experiments a conditional U-Net [70] architecture and the Adam optimizer [71] with PyTorchs OneCylce learning rate scheduler [72]. On MNIST we trained without exponential moving average (EMA) while on CIFAR10 we conducted experiments with and without EMA.

**Set up on MNIST**. We used an attention resolution of $[4, 2]$, 3 resnet blocks and a channel multiplication of $[1, 2, 2, 2, 2]$ and trained with a maximal learning rate of $10^{-4}$ for $50k$ iterations and a batch size of $1024$. For all MNIST training runs we used one A100 GPU per run, taking approximately 17 hours.

**Set up on CIFAR10**. We used an attention resolution of $[8]$, 4 resnet blocks and a channel multiplication of $[1, 2, 2, 2, 2]$. For the experiments without EMA, we used the same setup as with MNIST, but trained the models in parallel on two A100 GPUs for $300k$ iterations with an effective batch size of $1024$. When training with EMA, we followed the set up of Song et al. [16] using an EMA decay of $0.9999$ for all FVP dynamics and an EMA decay of $0.999$ for all FVE dynamics. In contrast to Song et al. [16] we used PyTorchs OneCycleLR learning rate scheduler with a maximal learning rate of $2 \cdot 10^{-4}$ and trained only for $1mio$ iterations instead of the $1.3mio$ iterations in Song et al. [16].

## D    Additional experiments

In addition to the experiments presented in the main part, we provide additional results here, including a full evaluation of FVE dynamics on MNIST, as well as training on CIFAR10 without EMA.

**Evaluation of different Hurst indices of FVE dynamics on MNIST**. In Table 4 we provide the evaluation of FVE dynamics. For the ease of comparisan, we include the quantitative results on FVP dynamics already presented in the main part. For FVE dynamics both, the super-diffusive regime and the sub-diffusive regime achieve a higher FID as the purely Brownian dynamics for $K = 1, 2$ throughout all tested Hurst indices and for $K = 3$ throughout all tested Hurst indices except for $H = 0.9$ with a higher pixel-wise diversity in the sub-diffusive regime of $H < 0.5$.

**Training on CIFAR10 without EMA**. As Song et al. [16] point out, the empirically optimal EMA decay rate for VP dynamics differs from that for VE dynamics. Since we do not have the computational resources to optimize the EMA decay rate for every configuration of our framework, we evaluated it in line with Song et al. [16] using a consistent EMA decay rate of $0.999$ across all configurations of FVE dynamics and $0.9999$ across all configurations of FVP dynamics. Nevertheless, because the optimal EMA decay rate appears to depend on the dynamics of the underlying stochastic

| MNIST | $H=0.9$ | | $H=0.7$ | | $H=0.5$ | | $H=0.1$ | |
|---|---|---|---|---|---|---|---|---|
| | FID↓ | VS$_p$↑ | FID↓ | VS$_p$↑ | FID↓ | VS$_p$↑ | FID↓ | VS$_p$↑ |
| **BM driven** | | | | | | | | |
| VP (retrained) | - | - | - | - | 1.44 | 23.64 | - | - |
| **MA-fBM driven** | | | | | | | | |
| FVP($H,K=1$) | 2.86 | 23.56 | 3.01 | 23.78 | 2.81 | 23.69 | 2.92 | 23.59 |
| FVP($H,K=2$) | 1.93 | 24.00 | 2.30 | 23.82 | 2.92 | 23.63 | 2.56 | 23.82 |
| FVP($H,K=3$) | **0.72** | 24.18 | 2.67 | 23.96 | 3.51 | 23.78 | 4.87 | 23.60 |
| FVP($H,K=4$) | **1.22** | **24.76** | **0.86** | **24.39** | 1.86 | **24.50** | 6.25 | 23.89 |
| FVP($H,K=5$) | 2.17 | **25.15** | 1.36 | 24.63 | 4.89 | **24.56** | 9.57 | 23.70 |

(a) Conditional image generation on MNIST with FVP.

| MNIST | $H=0.9$ | | $H=0.7$ | | $H=0.5$ | | $H=0.1$ | |
|---|---|---|---|---|---|---|---|---|
| | FID↓ | VS$_p$↑ | FID↓ | VS$_p$↑ | FID↓ | VS$_p$↑ | FID↓ | VS$_p$↑ |
| **BM driven** | | | | | | | | |
| VE (retrained) | - | - | - | - | 10.82 | 24.20 | - | - |
| **MA-fBM driven** | | | | | | | | |
| FVE($H,K=1$) | **10.06** | 24.05 | **9.95** | 24.24 | 10.30 | 24.22 | 9.98 | 24.20 |
| FVE($H,K=2$) | **9.82** | 24.07 | **9.73** | 24.13 | **9.89** | 24.15 | **9.42** | 24.28 |
| FVE($H,K=3$) | 11.02 | **24.53** | **9.96** | 24.37 | **9.74** | 24.42 | **10.12** | 24.44 |
| FVE($H,K=4$) | 31.67 | 22.44 | 11.37 | **24.34** | 11.25 | **24.54** | 9.56 | **24.58** |
| FVE($H,K=5$) | 50.42 | 23.74 | 22.03 | 22.09 | 25.51 | 23.08 | 10.39 | **24.33** |

(b) Conditional image generation on MNIST with FVE.

Table 4: FID and pixel-wise diversity scores of GFDM compared to the original setting of purely Brownian driven dynamics VE and VP. In bold the scores that are better than both purely Brownian driven dynamics VE and VP. The overall best scores within the experiment are boxed in.

| CIFAR10 | $H=0.9$ | | $H=0.5$ | | $H=0.1$ | |
|---|---|---|---|---|---|---|
| | FID↓ | VS$_p$↑ | FID↓ | VS$_p$↑ | FID↓ | VS$_p$↑ |
| **BM driven** | | | | | | |
| VE (retrained) | - | - | 9.38 | 3.21 | - | - |
| VP (retrained) | - | - | 17.29 | 2.24 | - | - |
| **MA-fBM driven** | | | | | | |
| FVE($H,K=1$) | 9.52 | **3.22** | 9.46 | **3.22** | **8.93** | **3.26** |
| FVE($H,K=2$) | **8.99** | **3.26** | 9.62 | **3.22** | 10.23 | 3.09 |
| FVE($H,K=3$) | 16.67 | 2.175 | 13.41 | 2.94 | 16.54 | 2.62 |
| FVE($H,K=4$) | 40.03 | 1.41 | 17.74 | 2.26 | 14.49 | 2.46 |

Table 5: Quantitative results for FVE dynamics and varying Hurst index on CIFAR10 trained without EMA. In bold the scores that are better than both purely Brownian driven dynamics VE and VP. The overall best scores within the experiment are boxed in.

process, we also evaluate our framework without EMA. In Table 5 we observe that the best performing configuration in terms of FID is FVP($H = 0.9, K = 2$) with an FID of $8.99$ and FVP($H = 0.1, K = 1$) with an FID of $8.93$ compared to the purely Brownian dynamics VE with an FID of $9.38$ and VP with an FID of $17.29$. Due to limited computational resources we only compared the best purely Brownian dynamics (VE) with the performance of corresponding augmented FVE dynamic of GFDM. As to be expected, using EMA for training of GFDM results in improved performnace w.r.t. image quality measured by FID obervable in Table 2b.

**Effect of the number of augmenting processes in the super-diffusive regime**. Additionally, in Figure 7, we show the FID evolution of the super-diffusive regime for various numbers of augmenting processes, showing a similar pattern that either that $K = 2$ or $K = 3$ yields the best performance across different datasets and dynamics.

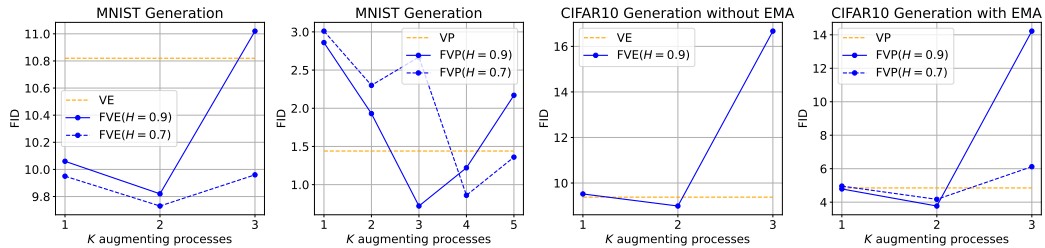

Figure 7: Dynamics driven by MA-fBM with super-diffusive Hurst index $H = 0.9$ and $K = 0.7$ perform in all four experiments we conducted better than the original purely Brownian driven dynamics, where either $K = 2$ or $K = 3$ yields the best performance.

| | FID↓ | | | |
|---|---|---|---|---|
| | 250 NFEs | 500 NFEs | 750 NFEs | 1000 NFEs |
| **BM driven** | | | | |
| VE | $5.65_{\pm 0.02}$ | $5.28_{\pm 0.04}$ | $5.20_{\pm 0.02}$ | $5.19_{\pm 0.02}$ |
| VP | $15.12_{\pm 0.11}$ | $5.86_{\pm 0.07}$ | $4.79_{\pm 0.11}$ | $4.79_{\pm 0.11}$ |
| **MA-fBM driven** | | | | |
| FVP($H=0.7, K=2$) | $15.44_{\pm 0.09}$ | $4.47_{\pm 0.03}$ | $4.13_{\pm 0.03}$ | $4.12_{\pm 0.03}$ |
| FVP($H=0.9, K=2$) | $12.44_{\pm 0.08}$ | $3.71_{\pm 0.03}$ | $\mathbf{3.70}_{\pm 0.03}$ | $\mathbf{3.70}_{\pm 0.03}$ |

Table 6: Averaged FID values for different NFEs of the super-diffusive regime compared to purely Brownian dynamics.

# E Illustration of generated data

**Visual comparison of generated CIFAR10 images sampled with SDE dynamics**.

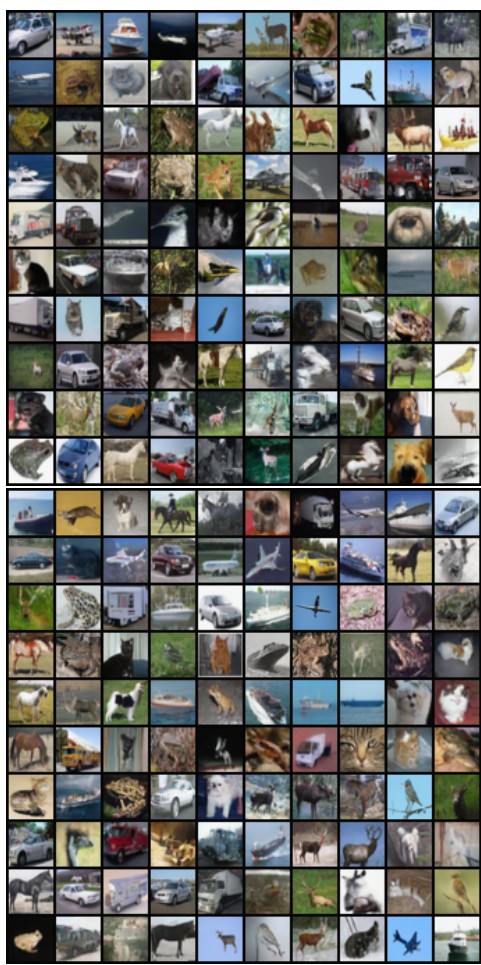 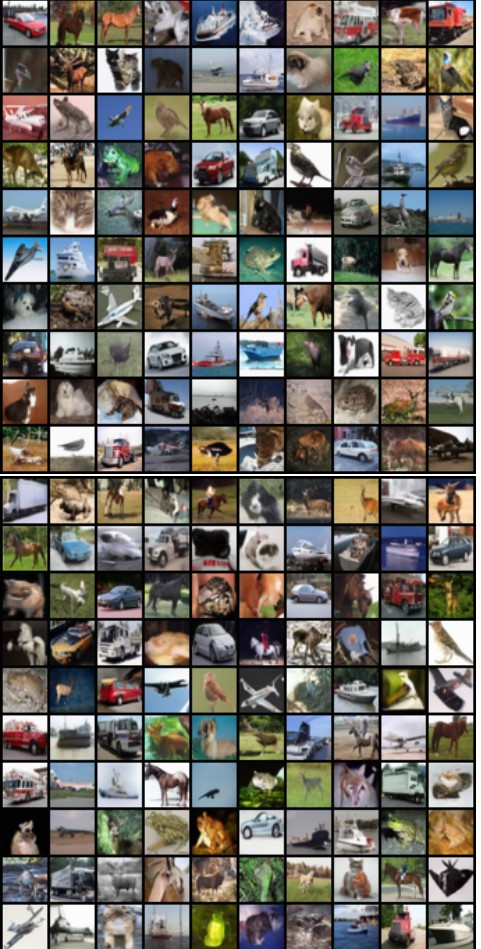

(a) Purely Brownian VP sample.
(b) Super-diffusive regime of MA-fBM with $H = 0.9$.

Figure 8: (LHS) Images generated with the purely Brownian driven VP dynamics sampled with SDE dynamics, a FID of $4.85$ and a pixel-wise diversity of $3.42$. (RHS) Images generated with $\text{FVP}(H = 0.9, K = 2)$ dynamics sampled with SDE, a FID of $3.77$ and a pixel-wise diversity of $3.60$.

**Visual comparison of generated CIFAR10 images sampled with PF ODE dynamics**.

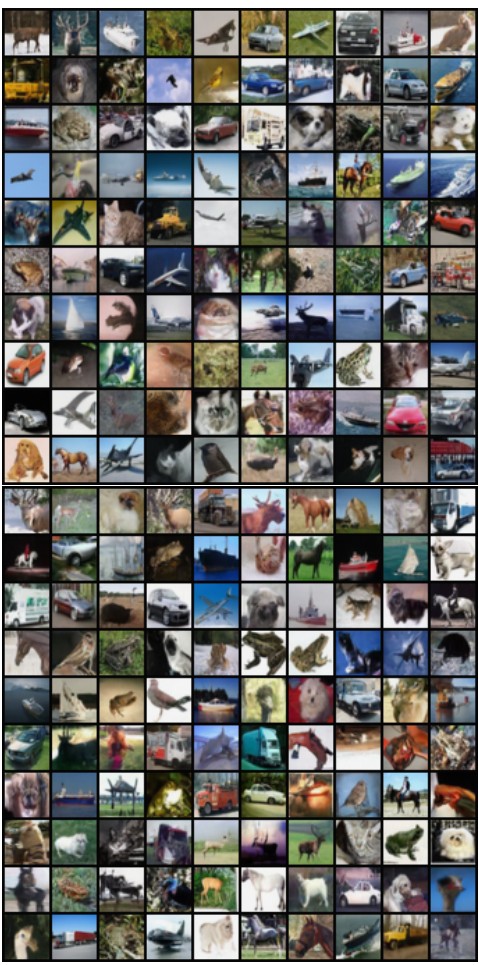 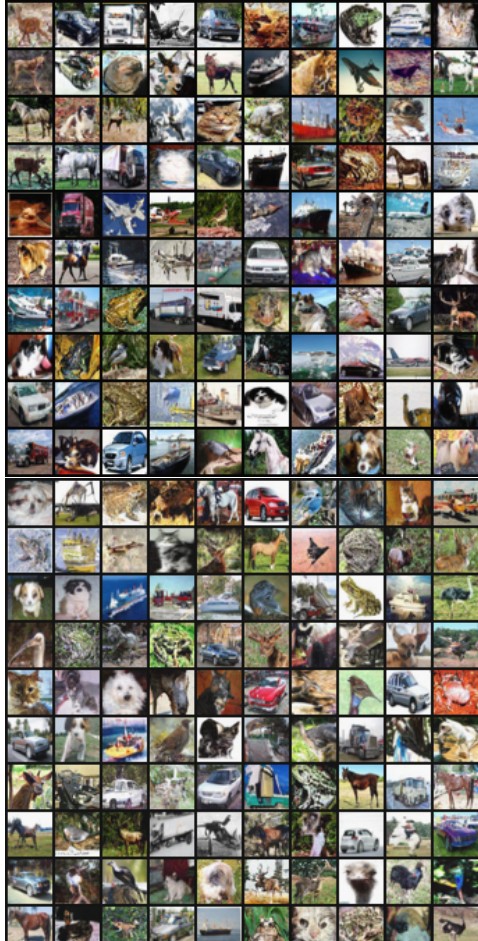

(a) Purely Brownian VP sample.  (b) Super-diffusive regime of MA-fBM with $H = 0.9$.

Figure 9: (RHS) Images generated with the purely Brownian driven VP dynamics sampled with PF ODE, a FID of $5.63$ and pixel-wise diversity of $3.91$. (LHS) Images generated with FVP($H = 0.9, K = 2$) dynamics sampled with PF ODE, a FID of $12.36$ and pixel-wise diversity of $4.89$.

**Visual comparison of generated MNIST images sampled from SDE**.

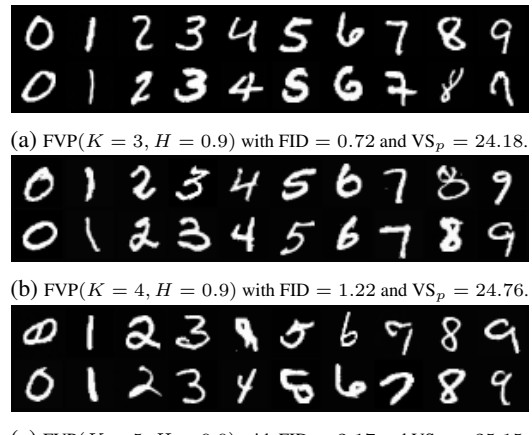

(a) FVP($K = 3, H = 0.9$) with FID $= 0.72$ and $\text{VS}_p = 24.18$.

(b) FVP($K = 4, H = 0.9$) with FID $= 1.22$ and $\text{VS}_p = 24.76$.

(c) FVP($K = 5, H = 0.9$) with FID $= 2.17$ and $\text{VS}_p = 25.15$.

Figure 10: Diversifying effect of the augmenting processes with FVP dynamics on MNIST. The super-diffusive regime with $H = 0.9$: For $K = 5$ instead of $K = 3$ augmenting processes the pixel VS increases from 24.18 to 25.15.

## F   Computational cost of augmenting processes

In this section we compare the computation time of GFDM to the purely Brownian setting of traditional diffusion models. For a given Hurst index $H \in (0, 1)$ and a given $K$, the optimal coefficients $\omega_1, ..., \omega_K$ are calculated only once before training. For completeness of our quantitative compute time evaluation, we provide the average computation time in seconds, needed to compute $\omega_1, ..., \omega_K$ on a GPU Tesla V100 with 32 GB RAM. We randomly sample 1000 times $H \sim \mathcal{U}[0.1, 0.9]$ for a given $K \in \{1, 2, 3, 4, 5\}$ and report the average computation time in Table 7.

|  | $K = 1$ | $K = 2$ | $K = 3$ | $K = 4$ | $K = 5$ |
|---|---|---|---|---|---|
| **time** [s] | 0.0043 | 0.0003 | 0.0003 | 0.0003 | 0.0003 |

Table 7: Averaged time in seconds needed before training to calculate for a given $K$ the optimal approximation coefficients using the approach of Daems et al. [27].

**Computation time during training** The computational difference during training consists of the computation of the covariance matrix $\Sigma_t$ instead of the marginal variance and sampling from a multivariate Gaussian instead of a univariate Gaussian. Note however, that we only need to calculate $\Sigma_t$ for $D = 1$ and also sample only once for a given time $t$ and a given data point. In Table 8 and Table 9 we report the average time of one training step measured in seconds calculated over 1000 training steps on CIFAR10. The underlying conditional U-Net has $58.7mio$ and EMA is applied. The batch size is 128 and all computation have been carried out on a GPU Tesla V100 with 32 GB RAM.

We observe that the computation time depends only minimaly increases when switching from the original model to the augmented system and increases across FVE and FVP dynamics by at most $11/1000$ seconds, while the choice of the Hurst index $H$ has no effect on the computation time.

**Computation time during sampling** Since the augmented system depends for fixed $K$ only on the approximating coefficients $\omega_1, ..., \omega_K$ it would suffice to report the average sampling time for FVP and FVE dynamics for varying $K$. Nevertheless, we report for $H \in \{0.9, 0.5, 0.1\}$ in Table 10 and Table 11 the average time to sample a batch of 1000 images over 1000 discretization steps of the reverse-time SDE over 10 trials. We observe that the average time in seconds for one sampling step in the reverse dynamics of FVE and FVP dynamics increases for $K \le 4$ by at most $2/100$ seconds. Only for $K = 5$ we observe a significant increase of average sampling time of roughly $4/10$ seconds.

| training step time [s] | $H = 0.9$ | $H = 0.5$ | $H = 0.1$ | average |
|---|---|---|---|---|
| VE | - | $0.0478_{\pm 0.1702}$ | - | 0.0478 |
| FVE$(H, K = 1)$ | $0.0489_{\pm 0.0927}$ | $0.0483_{\pm 0.0893}$ | $0.0485_{\pm 0.0922}$ | 0.0486 |
| FVE$(H, K = 2)$ | $0.0486_{\pm 0.0944}$ | $0.0484_{\pm 0.0484}$ | $0.0485_{\pm 0.0904}$ | 0.0485 |
| FVE$(H, K = 3)$ | $0.0487_{\pm 0.0967}$ | $0.0484_{\pm 0.0892}$ | $0.0493_{\pm 0.0924}$ | 0.0488 |
| FVE$(H, K = 4)$ | $0.0492_{\pm 0.0939}$ | $0.0484_{\pm 0.0897}$ | $0.0487_{\pm 0.0952}$ | 0,0488 |
| FVE$(H, K = 5)$ | $0.0487_{\pm 0.0939}$ | $0.0488_{\pm 0.0906}$ | $0.0486_{\pm 0.0933}$ | 0.0487 |

Table 8: Average time in seconds for one training step with FVE dynamics on CIFAR10 with a batch size of 128, a conditional U-Net with $58.7mio$ parameters and EMA.

| training step time [s] | $H = 0.9$ | $H = 0.5$ | $H = 0.1$ | average |
|---|---|---|---|---|
| VP | - | $0.0478_{\pm 0.1688}$ | - | 0.0478 |
| $K = 1$ | $0.0475_{\pm 0.0899}$ | $0.0475_{\pm 0.0938}$ | $0.0475_{\pm 0.0900}$ | 0.0475 |
| $K = 2$ | $0.0476_{\pm 0.0907}$ | $0.0477_{\pm 0.0917}$ | $0.0481_{\pm 0.0907}$ | 0.0478 |
| $K = 3$ | $0.0483_{\pm 0.0937}$ | $0.0477_{\pm 0.0909}$ | $0.0477_{\pm 0.0950}$ | 0.0479 |
| $K = 4$ | $0.0476_{\pm 0.0899}$ | $0.0479_{\pm 0.0916}$ | $0.0484_{\pm 0.0937}$ | 0.0480 |
| $K = 5$ | $0.0484_{\pm 0.0942}$ | $0.0479_{\pm 0.0925}$ | $0.0479_{\pm 0.0930}$ | 0.0481 |

Table 9: Average time in seconds for one training step with FVP dynamics on CIFAR10 with a batch size of 128, a conditional U-Net with $58.7mio$ parameters and EMA.

| sampling step time [s] | $H = 0.9$ | $H = 0.5$ | $H = 0.1$ | average |
|---|---|---|---|---|
| VE | - | $2.3092_{\pm 0.1462}$ | - | 2.3092 |
| $K = 1$ | $2.3125_{\pm 0.1275}$ | $2.3269_{\pm 0.1069}$ | $2.3261_{\pm 0.1093}$ | 2.3218 |
| $K = 2$ | $2.3095_{\pm 0.1280}$ | $2.3297_{\pm 0.1077}$ | $2.3107_{\pm 0.1593}$ | 2.3166 |
| $K = 3$ | $2.3071_{\pm 0.1213}$ | $2.3133_{\pm 0.1083}$ | $2.3063_{\pm 0.1058}$ | 2.3089 |
| $K = 4$ | $2.3322_{\pm 0.1086}$ | $2.3323_{\pm 0.1067}$ | $2.3156_{\pm 0.1122}$ | 2.3267 |
| $K = 5$ | $2.6515_{\pm 0.0930}$ | $2.6560_{\pm 0.1067}$ | $2.6510_{\pm 0.0953}$ | 2.6528 |

Table 10: Average time in seconds for one sampling step in the reverse dynamics of FVE to generate data of dimension $(3, 32, 32)$ with a batch size of 1000 using a conditional U-Net with $58.7mio$ and EMA.

| sampling step time [s] | $H = 0.9$ | $H = 0.5$ | $H = 0.1$ | average |
|---|---|---|---|---|
| VP | - | $2.3013_{\pm 0.1511}$ | - | 2.3013 |
| $K = 1$ | $2.3036_{\pm 0.1290}$ | $2.3120_{\pm 0.1062}$ | $2.3031_{\pm 0.1133}$ | 2.3062 |
| $K = 2$ | $2.3139_{\pm 0.1166}$ | $2.3070_{\pm 0.1102}$ | $2.3154_{\pm 0.1555}$ | 2.3121 |
| $K = 3$ | $2.3134_{\pm 0.1246}$ | $2.3168_{\pm 0.1056}$ | $2.3309_{\pm 0.1096}$ | 2.3204 |
| $K = 4$ | $2.3199_{\pm 0.1109}$ | $2.3091_{\pm 0.1132}$ | $2.3210_{\pm 0.1383}$ | 2.3167 |
| $K = 5$ | $2.6568_{\pm 0.0984}$ | $2.6603_{\pm 0.0978}$ | $2.6692_{\pm 0.0975}$ | 2.6621 |

Table 11: Average time in seconds for one sampling step in the reverse dynamics of FVP to generate data of dimension $(3, 32, 32)$ with a batch size of 1000 using a conditional U-Net with $58.7mio$ parameters and EMA.

## G  Likelihood computation

Given the approximate PF ODE corresponding to the augmented forward process

$$d\mathbf{z}_t = \underbrace{\left\{ \mathbf{F}(t)\mathbf{z}_t - \frac{1}{2}\mathbf{G}(t)\mathbf{G}(t)^T \left[ S_{\boldsymbol{\theta}}(\mathbf{z}_t, t) + \nabla_{\mathbf{z}} \log q_t(\mathbf{y}_t^{[K]}) \right] \right\}}_{:= \tilde{\mathbf{f}}_{\boldsymbol{\theta}}(\mathbf{z}_t, t)} dt, \quad t \in [0, T] \tag{120}$$

we estimate according to Song et al. [16] the log-likelihoods of test data $\mathbf{z}_0$ under the learned density $\tilde{p}_0^{aug}$ via

$$\log \tilde{p}_0^{aug}(\mathbf{z}_0) = \log \tilde{p}_T^{aug}(\mathbf{z}_T) + \int_0^T \nabla \tilde{\mathbf{f}}_{\boldsymbol{\theta}}(\mathbf{z}_t, t) dt. \tag{121}$$

According to Song et al. [16], we integrate over $[\epsilon, T]$ rather than $[0, T]$, using the same value of $\epsilon = 10^{-3}$, which has been empirically shown to yield the best performance when simulating the SDE. For $\epsilon \neq 0$ and type II fBM we need to adjust the starting value of the augmenting processes from zero to a jointly sampled vector $\mathbf{y}_\epsilon = (y_\epsilon^1, ..., y_\epsilon^K) \sim \mathcal{N}(\mathbf{0}_K, \mathbf{\Lambda}_\epsilon)$ with

$$(\mathbf{\Lambda}_\epsilon)_{k,l} = \mathbb{E}\left[ y_\epsilon^k y_\epsilon^l \right] = \int_0^\epsilon e^{-(\gamma_k + \gamma_l)(\epsilon - s)} ds = \frac{1 - e^{-(\gamma_k + \gamma_l)\epsilon}}{\gamma_k + \gamma_l}. \tag{122}$$

Using the exact likelihood of $\mathbf{y}_\epsilon$ and the independence of $\mathbf{y}_\epsilon$ and $\mathbf{x}_0$ we have

$$\log \tilde{p}_0^{aug}(\mathbf{z}_\epsilon) = \log \tilde{p}_0(\mathbf{x}_0) + \log q_\epsilon(\mathbf{y}_\epsilon) \tag{123}$$

where $\tilde{p}_0$ is the learned density of $\mathbf{x}_0$ corresponding to $\boldsymbol{\theta}$. Hence in total

$$\log \tilde{p}_0(\mathbf{x}_0) \overset{(121)}{=} \log \tilde{p}_T^{aug}(\mathbf{z}_T) + \int_0^T \nabla \tilde{\mathbf{f}}_{\boldsymbol{\theta}}(\mathbf{z}_t, t) dt - \log q_\epsilon(\mathbf{y}_\epsilon) \tag{124}$$

and we define the negative log-likelihoods $NLLs$ of test data $\mathbf{x}_0$ under the learned density by

$$NLLs(\mathbf{x}_0, \boldsymbol{\theta}) := -\log \tilde{p}_0^{aug}(\mathbf{z}_0) + \log q_\epsilon(\mathbf{y}_\epsilon). \tag{125}$$

## H  Challenges in the attempt to generalize

In this work, we seek to determine the extent to which the continuous-time framework of score-based generative models can be generalized from an underlying BM to an underlying fBM. For a fBM $W^H$ it is not straightforward to define the forward process

$$X_t = X_0 + \int_0^t f(X_s, s) ds + \int_0^t g(X_s, s) dW_s^H, \quad t \in [0, T] \tag{126}$$

driven by fBM, since fBM is neither a Markov process nor a semimartingale [25], and hence Itô calculus may not be applied, to define the second integral. However, a definition of the integral w.r.t. fBM is established [25, 73] such that the remaining problem is the derivation of the reverse-time model. Following the second and more intuitive derivation of the reverse-time model for BM from Anderson [32], the conditional backward Kolmogorov equation and the unconditional forward Kolmogorov equation are applied. Starting point of the derivation is to rewrite $p(x_t, t, x_s, s) = p(x_s, s | x_t, t) p(x_t, t)$ with Bayes theorem to calculate with the product rule

$$\frac{\partial p(x_t, t, x_s, s)}{\partial t} = \frac{\partial p(x_s, s | x_t, t)}{\partial t} p(x_t, t) + \frac{\partial p(x_t, t)}{\partial t} p(x_s, s | x_t, t), \quad s \geq t. \tag{127}$$

Replacing $\frac{\partial p(x_t, t)}{\partial t}$ with the RHS of the unconditional forward Kolmogorov equation and $\frac{\partial p(x_s, s | x_t, t)}{\partial t}$ with the RHS of the conditional backward Kolmogorov equation one derives an equation that only depends on the joint density $p(x_t, t, x_s, s)$. Using Bayes theorem again leads to a conditional backward Kolmogorov equation for $p(x_t, t | x_s, s)$ that defines the dynamics of the reverse process by the one-to-one correspondence between the conditional backward Kolmogorov equation and the reverse-time SDE [32]. Following these steps for fBM, starting from eq. (127) and deploying the

one-to-one correspondence of fBM and the evolution of its density [73], we could replace $\frac{\partial p(x_t,t)}{\partial t}$ in (127) by the RHS of

$$\frac{\partial p(x,t)}{\partial t} = \sum_{i=1}^{d} f_i(t,x) \frac{\partial p(t,x)}{\partial x_i} + H t^{2H-1} \sum_{i,j=1}^{d} g_{ij}(x,t) \frac{\partial^2 p(t,x)}{\partial x_i \partial x_j}. \tag{128}$$

The missing part is however an analogous to the conditional backward Kolmogorov equation to replace $\frac{\partial p(x_s,s|x_t,t)}{\partial t}$ in eq. (127). The derivation of such an equation is to the best of our knowledge yet unsolved problem and hence the limiting factor in the generalization of continuous-time score-based generative models from an underlying BM to an underlying fBM.

# I Notational conventions

| | |
|---|---|
| $[0, T]$ | Time horizon with terminal time $T > 0$ |
| $X = (X_t)_{t \in [0,T]}$ | Stochastic forward process taking values in $\mathbb{R}$ |
| $D \in \mathbb{N}$ | Data dimension |
| $\mathbf{X}$ | Vector valued stochastic forward process $\mathbf{X} = (\mathbf{X}_t)_{t \in [0,T]}$ with $\mathbf{X}_t = (X_{t,1}, ..., X_{t,D})$ |
| $\overline{\mathbf{X}}$ | Reverse time stochastic process with $\overline{\mathbf{X}}_t = \mathbf{X}_{T-t}$ |
| $\mathbf{f}$ | Vector valued function $\mathbf{f} : \mathbb{R}^D \times [0, T] \to \mathbb{R}^D$ |
| $\mu, g$ | Functions $\mu, g : [0, T] \to \mathbb{R}$ |
| $\overline{\mathbf{f}}$ | Reverse time function with $\overline{\mathbf{f}}(\mathbf{x}, t) = \mathbf{f}(\mathbf{x}, T - t)$ |
| $\bar{\mu}, \bar{g}$ | Reverse time functions with $\bar{\mu}(t) = \mu(T - t)$ and $\bar{g}(t) = g(T - t)$ |
| $p_0$ | Data distribution |
| $p_t$ | Marginal density of (augmented) forward process at $t \in [0, T]$ |
| $B$ | Brownian motion (BM) |
| $H$ | Hurst index $H \in (0, 1)$ |
| $W^H$ | Type I fractional Brownian motion (fBM) |
| $B^H$ | Type II fractional Brownian motion (fBM) |
| $Y^\gamma = (Y_t^\gamma)_{t \in [0,T]}$ | Ornstein–Uhlenbeck (OU) process with speed of mean reversion $\gamma \in \mathbb{R}$ |
| $K \in \mathbb{N}$ | Number of augmenting processes |
| $\gamma_1, ..., \gamma_K$ | Geometrically spaced grid |
| $\omega_1, ..., \omega_K$ | Approximation coefficients |
| $\boldsymbol{\omega}$ | Optimal approximation coefficients $\boldsymbol{\omega} = (\omega_1^\star, ..., \omega_K^\star)$ |
| $\bar{\omega}$ | Sum of optimal approximation coefficients |
| $\hat{B}^H$ | Markov-approximate fractional Brownian motion (MA-fBM) |
| $k$ | $k \in \mathbb{N}$ with $1 \leq k \leq K$ |
| $Y^k$ | OU processes $Y^k = Y^{\gamma_k}$ |
| $\mathbf{Y}^1, ..., \mathbf{Y}^K$ | Augmenting processes with $\mathbf{Y}^k = (Y^k, ..., Y^k)$ |
| $\mathbf{F}, \mathbf{G}$ | Vector valued functions $\mathbf{F}, \mathbf{G} : [0, T] \to \mathbb{R}^{D \cdot (K+1)}$ |
| $\overline{\mathbf{F}}, \overline{\mathbf{G}}$ | Reverse time vector valued functions with $\overline{\mathbf{F}}(t) = \mathbf{F}(T - t)$ and $\overline{\mathbf{G}}(t) = \mathbf{G}(T - t)$ |
| $\mathbf{Z}$ | By $\mathbf{Y}^1, ..., \mathbf{Y}^K$ augmented forward process |
| $\mathbf{Y}^{[K]}$ | Stacked vector of augmenting processes |
| $q_t$ | Marginal density of $\mathbf{Y}^{[K]}$ at $t \in [0, T]$ |
| $\boldsymbol{\theta}$ | Weight vector of a neural network |

