# OpenReview forum: "Generative Fractional Diffusion Models"
_NeurIPS.cc/2024/Conference — NeurIPS 2024 poster_

### Official Review · Reviewer_kZkj · 2024-07-08

**Soundness:** 2
**Presentation:** 3
**Contribution:** 3
**Rating:** 6
**Confidence:** 3

**Summary:**

The authors introduce a new type of continuous score-based generative models relying on fractional diffusion, a type of diffusion where Brownian Motion BM is replaced by its fractional counterpart fBM, where noise increments or either positively correlated (Hurst index $1 > H > 1/2$) or negatively correlated ($0 < H < 1/2$). The authors use a Markovian approximation of fBM, consisting in the addition of $K$ correlated OU processes as the underlying dynamics guiding the diffusion. This lets them obtain tractable learning and inference, using the augmented process (noised data, OU processes) and the corresponding score. The augmented score matching loss introduced makes it possible to use a single D-dimensional score model to optimally approximate the full score in $\mathbb{L}_2$, where D is the data dimensionality. The authors then design a set of experiments on MNIST and CIFAR10 to validate the performance of the new method and the effect of varying the hyper-parameters ($K, H$), and outline its advantage over classical score-based diffusion.

**Strengths:**

In terms of theory, the paper is quite satisfying. It is well contextualised in the field of research of fractional Brownian Motion, and leverages a good set of techniques to achieve its goals, and overcome the limitations of previous similar approaches with this noise regime. These aspects are treated efficiently and with clarity.

- Tractable learning and inference
- D-dimensional score model even though one has to consider the $D\cdot (K+1)$-dimensional augmented process
- Marginal statistics for faster training, and explicit formulaes for the VE noise schedule

**Weaknesses:**

While the paper tackles the extension of diffusion to fractional Brownian motion very satisfyingly, it seems to me that it suffers from an insufficient and unclear empirical study.
- Performance gains are mildly convincing and the effect of the varying hyper-parameters are not that clear. The general idea seems to say that choosing $H>1/2$ and $K=3$ yields good results, with higher $H$ providing some smoothing beneficial to learning the diffusion and higher $K$ bringing in more diversity, but to be honest I am not convinced by the interpretation of the results and the general setup.
- It could be beneficial to include a more exhaustive comparison with other approaches for diffusion with different noise regimes, and the different advantages/limitations, especially since the mentioned limitations for classical diffusion models (slow convergence, mode-collapse on imbalanced data, and lack of diversity) are not properly tackled. See, e.g., [1] [2], and even consider some more detailed experimental comparison with [3].


[1] Eliya Nachmani, Robin San Roman, and Lior Wolf. Denoising diffusion gamma models, 2021

[2] Jacob Deasy, Nikola Simidjievski, and Pietro Liò. Heavy-tailed denoising score matching, 2022.

[3] Eunbi Yoon, Keehun Park, Sungwoong Kim, and Sungbin Lim. Score-based generative models with Lévy Processes. In Thirty-seventh Conference on Neural Information Processing Systems, 2023

**Questions:**

Overall I am willing to raise my score if my concerns for the empirical study are addressed. Here are some additional questions and remarks.

- With $K \gg 1$ performance deteriorates. Does it means true fractional dynamics does not work well? What is happening? How does $K$ impact the approximation of true fBM (you cite a reference on Line 184, can you give something quantitative?)

- Why for $H=1/2$ do we see improvement with the augmented dynamics? Because of the correlation structure?

- Table 1 and effect of $K$ for $H = 1/2$: Not much difference, until FID deteriorates, where then $\textrm{VS}_p$ increases. If the sample quality heavily deteriorates and that the model produces random out of distribution samples, then it is not a very interesting situation and the metric loses meaning. Moreover no discussion on why choosing too high of a $K$ eventually leads to bad samples. Numerical instabilities? Training loss? Difficulty learning the augmented process?

- The Vendi score thus does not seem to be such a good choice as we cannot understand the tradeoff in quality and diversity. It would be better to use some precision/recall metric (or density/coverage) where diversity relates to covering the true data distribution. One could devise some summary statistic like an associated $F_1$ score to appreciate the tradeoff and see if a better 'Pareto optima' is attained.

- It is also not clear what is the effect of $H$ on the samples. It seems rough path $H < 1/2$ have larger jumps but the authors advocate for $H>1/2$ and on line 284 invoke heavy-tailedness. And what is meant by heavy-tailedness here, as the variance is finite?

- The effect of varying the hyperparameters does not seem very consistent. Will the observed remarks for $(K, H)$ hold with different datasets? Will this scale? It would also be nice to have multiple runs with the associated standard deviation to assess statistical significance here.

- In the abstract, limitations of diffusion models are mentioned, in particular (i) slow convergence and (ii) mode-collapse on imbalanced data, which are not addressed in the experiments and discussed properly. For (i), there is no mention of the number of diffusion steps/number of function evaluation in the experiments. For (ii), maybe try a run on CIFAR10\_LT.

- Table 3: no discussion on why the ODE has bad performance.

- Line 259: 'we evaluate GFDM on [..] test distribution coverage': this seems not to be done anywhere, as Vendi score does not relate to coverage in this sense?

- The diffusion is run on $[\epsilon, T]$ but the value of $\epsilon$ is nowhere to be found it seems, and no experiment to see its effect on  GFDM's performance?

- Line 32-34: why would BM's lack of control over sampled trajectories matter? The correlation structure is independent of the model, depending on the data it could also increase mixing time or actually make it harder to learn the score. Is it because the augmented process makes it possible for the model to incorporate better correlation structure? What are the basis of these claims?

- Line 83-84: $\hat f$ seems not to be properly introduced

- Line 168-169, eq (9): typo in the denominator of the last fraction?

- Line 210: $\Sigma_t$ seems not to be properly introduced

- Line 255-258: Why would using EMA on the models interfere with the SDE dynamics?

- Line 282: Did you not mean SOTA FID of 0.72, K = 3 and H = 0.9? What are you basing your subsequent interpretation of easier to learn smoother sample paths on?

- Line 291: Did not you mean super-diffusion here with $H = 0.9 > 1/2$? idem in caption of Figure 2, with $H=0.7$?

**Limitations:**

Properly addressed.

---

> ### Author Rebuttal · Authors · 2024-08-07
>
> Dear Reviewer kZkj,
>
> Thanks for the insightful comments, valuable feedback and very precise and thorough engagement with our work. See below for detailed answers to your questions and concerns:
>
> >Performance gains are mildly convincing and the effect of the varying hyper-parameters are not that clear. The general idea seems to say that choosing $H>1/2$ and $K=3$ yields good results, with higher $H$ providing some smoothing beneficial to learning the diffusion and higher $K$ bringing in more diversity, but to be honest I am not convinced by the interpretation of the results and the general setup.
>
> Inline with the reviewer's feedback, we now provide a more interpretable evidence. To this end, we retrained on CIFAR10 with all the configurations we investigated on MNIST before to better detect similarities. We find that on both datasets $H=0.9$ yields the best results with $K=3$ and MNIST and $K=2$ on CIFAR10, directly followed by $H=0.7$ with $K=4$ on MNIST and the $K=2$ on CIFAR10. The super-diffusive regime performs on MNIST $K\geq 3$ in $4$ out of $6$ configurations better than both purely Brownian driven baseline dynamics and on CIFAR10 for $3$ out of $4$ configurations with $K\leq2$. The attached Figure 2 indicates that there are dataset and dynamics specific configuration clusters, where GFDM performs better than the purely Brownian driven models.
>
> >Why for $H=1/2$ do we see improvement with the augmented dynamics? Because of the correlation structure?
>
> Yes, our hypothesis is that the known part of the score function is guiding the data generating process towards the direction of the true data distribution, since the starting distribution of the augmenting processes is known and all processes are driven by the same random path realization of Brownian motion. In the forward process $\mathbf{Y}_t$ does not depend on $\mathbf{X}_t$, but in the reverse model $\mathbf{Y}_t$ does depend on $\mathbf{X}_t$ since it depends on every entry on the score model evaluation $s_{\theta}(\mathbf{X}_t,t)$. Assume that $\mathbf{Y}_t$ does not leave its true reversed trajectory, in this case the trajectory of $\mathbf{Y}_t$ might serve as a corrector for the trajectory of $\mathbf{X}_t$.
>
> >The diffusion is run on $[\epsilon, T]$ but the value of $\epsilon$ is nowhere to be found it seems, and no experiment to see its effect on GFDM's performance?
>
> Thanks for noting. We defined $\epsilon$ in the Appendix (lines 730-731), but the definition should, of course, be in the main part of the paper. We have added this.
>
> >Line 83-84: $\hat f$ seems not to be properly introduced
>
> Thanks for the detailed review. Since we have $\mathbf{f}(\mathbf{x},\cdot):[0,T]\to\mathbb{R}^{D}$ for fixed $\mathbf{x}\in\mathbb{R}^{D}$ we intended to define $\mathbf{\overline{f}}(\mathbf{x},t):=\mathbf{f}(\mathbf{x}_t,T-t)$
> as well in line 80-81:
>
> Whenever $\mathbf{X}=(\mathbf{X}_{t})_{t\in[0,T]}$ is a stochastic process and $g$ is a function on $[0,T]$, we write $\mathbf{X}_{t}=\mathbf{X}_{T-t}$ for the reverse time model and $\bar{g}(t)=g(T-t)$ for the reverse time function.
>
> To make it more clear, we replaced line 80-81 by:
>
> Whenever $\mathbf{P}=(\mathbf{P}_{t})_{t\in[0,T]}$ is a stochastic process and $f$ is a function on $[0,T]$, we write $\mathbf{P}_{t}=\mathbf{P}_{T-t}$ for the reverse time model and $\bar{f}(t)=f(T-t)$ for the reverse time function.}
>
> >Line 168-169, eq (9): typo in the denominator of the last fraction?
>
> Thanks or finding this typo, which we incorporated.
>
> >Line 255-258: Why would using EMA on the models interfere with the SDE dynamics?
>
> Song et. al 2021 point out at that "For models trained with VE perturbations, we notice that $0.999$ works better than $0.9999$, whereas for models trained with VP perturbations it is the opposite. We therefore use an EMA rate of $0.999$ and $0.9999$ for VE and VP models respectively." Due to this empirical observation, the EMA decay rate seems to have a different effect on different underlying dynamics. Since we do not have enough compute to investigate the optimal EMA decay rate for every configuration $(H,K)$ we investigated instead on CIFAR10 both settings, training with and without EMA.
>
> > Line 282: Did you not mean SOTA FID of 0.72, K = 3 and H = 0.9? What are you basing your subsequent interpretation of easier to learn smoother sample paths on?
>
> This is absolutely right, thank you for noting. Our hypothesis here is that we have a smaller discretization error from continuous to discrete time, when the distribution transforming path is smoother. This is based on the observation that the discretization error of approximating the (rough path driven) SDE with Euler-Maruyama method results in a discretization error of order $N^{-\frac{1}{2}}$, while the discretization error of the Euler-method to simulate the (smooth path $t\mapsto t$ driven ) ODE is of order $N^{-1}$. We describe this in more detail in lines 95-112. We have changed the paragraph in lines 280-284 and rephrased it as a conjecture:
>
> By varying the Hurst index we observe in Table 2a that $H>0.5$ with FVP dynamics performs better in terms of FID achieving a SOTA FID of $0.72$ for FVP with $(H,K)=(0.9,3)$. Moreover, the configurations $(H,K)=(0.9,4),(0.7,4),(0.7,5)$ in the regime of $H>0.5$ perform all better than the purely Brownian dynamics VE and VP. We conjecture that this is due to the super-diffusion regime, smoothing the sample paths, making the dynamics easier to learn. The augmenting processes increase the pixel-wise diversity in terms of $VS_p$ and $VS^{min}_p$ in Table 2a as well for $H\in\{0.9,0.7\}$ compared to the original VP dynamics, again at the cost of a higher NLLs for more augmenting processes.

---

> ### Author Response · Authors · 2024-08-08
>
> Dear Reviewer kZkj,
>
> We sincerely apologize for the formatting issues in our previous response and the missing reference. Here is the corrected version, along with additional answers to the questions we couldn't cover previously due to the character limit:
>
> [1] Yang Song and Jascha Sohl-Dickstein and Diederik P Kingma and Abhishek Kumar and Stefano Ermon and Ben Poole. Score-Based Generative Modeling through Stochastic Differential Equations, International Conference on Learning Representations, 2021.
>
> >Why for do we see improvement with the augmented dynamics? Because of the correlation structure?
>
> Yes, our hypothesis is that the known part of the score function is guiding the data generating process towards the direction of the true data distribution, since the starting distribution of the augmenting processes is known and all processes are driven by the same random path realization of Brownian motion. In the forward process $\mathbf{Y}_t$ does not depend on $\mathbf{X}_t$, but in the reverse model $\mathbf{Y}_t$ does depend on $\mathbf{X}_t$, since it depends in every entry on the score model evaluation. Assume that $\mathbf{Y}_t$  does not leave its true trajectory when we simulate the reverse time model, in this case the trajectory of $\mathbf{Y}_t$ might serve as a corrector for the trajectory of $\mathbf{X}_t$.
>
> >With $K>>1$ the performance deteriorates. Does it means true fractional dynamics does not work well? What is happening?
>
> We believe that the most important factor for chosing the optimal number of augmenting processes is the dataset. In Table 1 of the attached Rebuttal PDF, we observe that on MNIST, $K=4,5$ works well in the super-diffusive regime, while on CIFAR10, the performance degrades in the same regime. In Figure 2 of the attached rebuttal PDF, we observe that for the same FVP dynamics, there are different well-performing regimes for the number of processes. (K=3,4,5 on MNIST and K=1,2 on CIFAR10).
>
> >How does $K$ impact the approximation of true fBM (you cite a reference on Line 184, can you give something quantitative?)
>
> The integrated $L_2$ error over time $[0,T]$ for different choices of $K$ is visualized in Figure 7 of Daems et. al[2]. The higher the value of $K$, the better the approximation. However, after a certain number of augmenting processes, the error saturates, depending on $H$.
>
> [2] Rembert Daems, Manfred Opper, Guillaume Crevecoeur, Tolga Birdal. Variational Inference for SDEs Driven by Fractional Noise. The Twelfth International Conference on Learning, 2024.
>
> > Table 1 and effect of $K$ for $H=1/2$: Not much difference, until FID deteriorates, where then increases. If the sample quality heavily deteriorates and that the model produces random out of distribution samples, then it is not a very interesting situation and the metric loses meaning. Moreover no discussion on why choosing too high of a eventually leads to bad samples. Numerical instabilities? Training loss? Difficulty learning the augmented process
>
> We agree with the reviewers point that a low-quality image with increased pixel-wise diversity is not a favorable situation. In Table 1 of the attached rebuttal PDF, we observe that the pixel-wise diversity increases with a higher number of augmenting processes on MNIST and CIFAR10. However, the quality seems to depend on the Hurst index and the dataset, as we observe on MNIST for larger $K$:
>
>
> | K | H=0.9 | H=0.7 |H=0.5 |H=0.1|
> |---|-------|-------|------|-----|
> | 4 | 1.22  | 0.86  |1.86  |6.25 |
> | 5 | 2.17  |  1.36 |4.89  |9.57 |
>
> while on CIFAR10 we observe a clear degradation in quality:
>
> | K | H=0.9 | H=0.7 |H=0.5 |H=0.1|
> |---|-------|-------|------|-----|
> | 4 | 29.72  | 8.45  |8.85  |5.02 |
> | 5 | 69.06  | 35.91 |96.54  |7.38 |
>
> Our hypothesis here is that $K$ controls the pixel-wise diversity, while the optimal number of augmenting processes for quality depends on the dataset and the dynamics (FVE or FVP).

---

> ### Author Response · Authors · 2024-08-08
>
> >The effect of varying the hyperparameters does not seem very consistent. Will the observed remarks for $(K,H)$ hold with different datasets? Will this scale? It would also be nice to have multiple runs with the associated standard deviation to assess statistical significance here.
>
> We agree with the reviewer's point that our previous presentation of our findings was not very clear. We hope to adress this issue in our attached rebuttal PDF:
>
> * in Table 1 we observe that the super-diffusive regime $(H>0.5)$ yields the best performance on MNIST and CIFAR10 in terms of quality and an increasing number of augmenting processes yields higher pixel-wise diversity.
> * in Figure 1 we observe that the quality evolution w.r.t. the number of augmenting processes evolves similar across dynamics and dataset.
>
> We also agree with the reviewer that it would be favorable to report the associated standard deviation across different run. However, this is beyond our computational resources. One training on CIFAR with 2 GPUs (Ampere A100 40 GB RAM) takes roughly 50 hours, making it unfeasible to train multiple times. Following the evaluation setup of Song et. al[1] we trained for 1 million iterations on CIFAR10 and evaluated one checkpoint every 100k iterations, to ensure that the result is not affected by overfitting. This ensures to some extend that the FIDs we report on CIFAR10 are not a lucky pick. Nevertheless, to adress the reviewers concerns we report in Figure 3c) the average FID and the associated standard deviation for a certain number of sampling steps over three rounds of sampling. Please note that to report one of these scores for $NFE=1000$, we need to sample $50k$ images on a single GPU (Ampere A100 with 40 GB RAM), which takes 16 hours.
>
> >In the abstract, limitations of diffusion models are mentioned, in particular (i) [...]. For (i), there is no mention of the number of diffusion steps/number of function evaluation in the experiments
>
> We thank the reviewer for pointing out that we did not mention the number of function evaluations (NFE). To address the reviewer's point, we compare the performance of the super-diffusive regime to the purely Brownian-driven dynamics in Table 1 of the attached rebuttal PDF. Evaluating the performance for different numbers of NFE of sampling steps in Figure 3a), we observe that the super-diffusive regime of MA-fBM saturates at $500$ NFE on a lower level than the purely Brownian driven dynamics.
>
> >(ii) mode-collapse on imbalanced data, which are not addressed in the experiments and discussed properly.[...] For (ii), maybe try a run on CIFAR10_LT.
>
> Unfortunately, this was not feasible given the short time period from review release to rebuttal. Based on the reviewer's suggestion, we will consider experiments on CIFAR10_LT in our future work.
>
> >Line 259: 'we evaluate GFDM on [..] test distribution coverage': this seems not to be done anywhere, as Vendi score does not relate to coverage in this sense?
>
> Our evaluation of negative log likelihood (NLL) in Table 1 and Table 2 was intendet to measure test distribution coverage, since NLL estimates the negative log-likelihood of test data under the learned density.
>
> >The diffusion is run on $[\epsilon, T]$ but the value of $\epsilon$ is nowhere to be found it seems, and no experiment to see its effect on GFDM's performance?
>
> Thanks for noting. We defined $\epsilon$ in the Appendix (lines 730-731), but the definition should, of course, be in the main part of the paper. We used throughout all experiments $\epsilon=1e-5$ for training and $\epsilon=1e-3$ for sampling according to Song et. al[1].
>
> >Line 32-34: why would BM's lack of control over sampled trajectories matter? The correlation structure is independent of the model, depending on the data it could also increase mixing time or actually make it harder to learn the score. Is it because the augmented process makes it possible for the model to incorporate better correlation structure? What are the basis of these claims?
>
> The Hurst index $H$ in our framework provides control over the diffusion processes trajectory in terms of roughness vs. mild randomness. Our results indicate that the more regular and correlated trajectories are preferred, which is not possible to achieve with BM. As we pointed out in our answer above, a smoother trajectory might decrease the discretization error from continuous time to discrete time steps.

---

> ### Author Response · Authors · 2024-08-08
>
> >Line 83-84: $\hat f$ seems not to be properly introduced
>
> Thanks for the very detailed review. Since we have $\mathbf{f}(\mathbf{x},\cdot):[0,T]\to\mathbb{R}^{D}$ for fixed $\mathbf{x}\in\mathbb{R}^{D}$ we intended to apply the definition for  $\overline{g}$ as well to $\mathbf{\overline{f}}(\mathbf{x},t):=\mathbf{f}(\mathbf{x}_t,T-t)$ in line 80-81. In order make it more clear that the notation applies to any process $\mathbf{P}$ and any function $f$ we replace line 80-81 by:
>
> "Whenever $\mathbf{P}=(\mathbf{P}_t) _{t\in[0,T]}$ is a stochastic process and $f$ is a function on $[0,T]$, we write $\mathbf{\overline{P}} _{t}=\mathbf{P} _{T-t}$ for the reverse time process and $\overline{f}(t)=f(T-t)$ for the reverse time function."
>
> >Line 168-169, eq (9): typo in the denominator of the last fraction?
>
> Thanks for finding this typo. We cut of the lower incomplete gamma function in the second part of $\mathbf{b}_{k}$. We replaced it by the correct term:
>
> $\mathbf{b} _{k} := \frac{T}{\gamma^{H+1/2} _{k}}P(H+1/2,\gamma _{k}T) - \frac{H+1/2}{\gamma _{k}^{H+3/2}}P(H+3/2,\gamma _{k}T)$
>
> >Line 210: $\Sigma_t$ seems not to be properly introduced
>
> Thanks for worrying about the details. In this case we already defined it in  line 195 - 198: "The missing components in the conditional covariance matrix $\Sigma_t$ of the augmented forward process are the conditional marginal variance of the forward process and the conditional marginal correlation between the forward process and the augmenting processes."
>
> >Line 291: Did not you mean super-diffusion here with $H = 0.9 > 1/2$? idem in caption of Figure 2, with $H=0.7$?
>
> Yes, thanks for noting. This is a typo and should be super-diffusion.

---

> > ### Author Response · Authors · 2024-08-08
> >
> > We truly appreciate the reviewers detailed engagement with our work and thank the reviewer for the valuable feedback.
> >
> > We hope that we have sufficiently addressed the weaknesses and questions raised by the reviewer.
> >
> > With best regards,
> > The Authors

---

> > > ### Comment · Reviewer_kZkj · 2024-08-09
> > >
> > > I would like to immensely thank the authors for their dedication in answering my questions in such details, and for the additional experiments they have run to address the issues that were raised.
> > >
> > > I acknowledge their answer, and will now reflect and discuss with the other reviewers before taking a final decision.
> > >
> > > With best regards

---

### Official Review · Reviewer_HxVk · 2024-07-12

**Soundness:** 2
**Presentation:** 3
**Contribution:** 3
**Rating:** 6
**Confidence:** 4

**Summary:**

The work proposes a theoretical framework for training score-based diffusion models based on fractional brownian motion.

Authors provide an approximation framework where fBM being approximated by Markovian noise to derive score-matching in Markov approximation.

Theoretical derivation is supplemented by experiments showing some gains w.r.t. score-matching driven by standard brownian motion.

**Strengths:**

An interesting idea of extending score-matching to non-Markovian setting and approximating otherwise intractable fractional BM with colored brownian motions.

**Weaknesses:**

It feels that motivation for why we should use fBM is not well elaborated.

Many different settings evaluated without sensitivity analysis of $H$ and $K$ on resulting quality. It is rather hard to interpret the findings whether the gains are from fBM or from multiple comparisons.

**Questions:**

If the process driven by fBM is adapted, can we use Gyöngy theorem to reduce the process to BM one and connect this score-matching procedure to BM driven score matching? If so, would it imply that gains are coming from having different path-integrator, rather than from just using fBM?

Can authors add pvalues to the tables?

**Limitations:**

Yes

---

> ### Author Rebuttal · Authors · 2024-08-07
>
> Dear Reviewer HxVk,
>
> We thank the reviewer for their valuable feedback.  Our detailed responses follow:
>
> > It feels that motivation for why we should use fBM is not well elaborated
>
> Having control over the diffusion trajectories are crucial for obtaining a trade-off between generation quality and diversity. Moreover, there is no true support for why the noise increments should be independent. Our framework provides two hyperparameters, the number of processes and the Hurst index to have control over diffusion processes and characterize the nature of trajectories, \emph{e.g.} roughness vs. mild randomness. Our results indicate that in both aspects, quality and diversity, the diffusion models driven by Brownian motion lead to suboptimal results and the best outcome are obtained when $K\geq 2$ and $H>0.5$. We now mention this in the introduction of our main paper.
>
> > Many different settings evaluated without sensitivity analysis of $H$ and $K$ on resulting quality. It is rather hard to interpret the findings whether the gains are from fBM or from multiple comparisons.
>
> We agree that the tables in our main paper could appear cluttered.
> To foster comparability of our experiments, we retrained our model on CIFAR10 on all configurations that we already investigated on MNIST and replace Table 1 and Table 2 in our paper with the attached Table 1. Comparing the two tables, we see a better performance of the super-diffusive regime ($H>0.5$) accross the two datasets, reaching in both experiemnts the best FID for $H=0.9$, with either $3$ or $2$ augmenting processes. In Figure 3 you see that the best performance in terms of FID of the purely Brownian dynamics is dominated by the super-diffusive regime of our method, saturating on a lower FID level with aleary $500$ NFE. In addition, the pixel-wise diversity of the best performing configuration on CIFAR is higher than the pixel-wise diversity of the two baseline dynamics.
>
> > If the process driven by fBM is adapted, can we use Gyöngy theorem to reduce the process to BM one and connect this score-matching procedure to BM driven score matching? If so, would it imply that gains are coming from having different path-integrator, rather than from just using fBM?
>
> Thanks for this very insightful question. Fractional Brownian motion does not fit into the setting of Gyöngy's theorem directly since it is not a semimartingale so that the forward process is not given in terms of a stochastic differential equation driven by a Brownian motion. The lack of the semimartingale property also makes the question of whether it is possible to match the marginal distributions of the forward process a formidable one which is far beyond the scope of this paper. Importantly, replacing fractional Brownian motion by an alternative Brownian motion model with the same marginal distributions would result in an entirely different dependence structure of the marginals of the forward process, hence possibly leading to entirely different results. This would certainly be very interesting to check but, unfortunately, once again beyond the scope of the article.
>
> > Can authors add pvalues to the tables?
>
> Unfortunately this is beyond our computational resources. One training on CIFAR with 2 GPUs Ampere A100 40 GB RAM takes roughly 50 hours, making it unfeasible to train multiple times. Following the evaluation set up of Song et. al[1] and Lou et. al[2] with a smaller number of trainable parameters we trained for 1 million iterations on CIFAR10 and evaluated one checkpoint every 100k iterations, to ensure that the result is not affected by overfitting. This ensures to some extend that the FIDs we report on CIFAR10 are not a lucky pick.
> Nevertheless, we report for the best four configuration in Figure 3 a) the average FID over three rounds of training with the corresponding standard deviation. Please note that we need to sample $3 x 50k$ samples to report one of these scores  with $1000$ NFE which takes 1 GPU  Ampere A100 with 40 GB RAM approximately 16 hours.
>
> With best regards,
> The Authors

---

> > ### Comment · Reviewer_HxVk · 2024-08-12
> >
> > Thank for the reply, I'll increase my score

---

> ### Author Response · Authors · 2024-08-08
>
> Dear Reviewer HxVk,
>
> We sincerely apologize for the the missing references:
>
> [1] Yang Song and Jascha Sohl-Dickstein and Diederik P Kingma and Abhishek Kumar and Stefano Ermon and Ben Poole. Score-Based Generative Modeling through Stochastic Differential Equations, International Conference on Learning Representations, 2021.
>
> [2] Aaron Lou and Stefano Ermon. Reflected Diffusion Models, International Conference on Machine Learning, 2023.
>
> We hope that we have sufficiently addressed the weaknesses and questions raised by the reviewer.
>
> With best regards,
> The Authors

---

### Official Review · Reviewer_kgmF · 2024-07-13

**Soundness:** 1
**Presentation:** 2
**Contribution:** 1
**Rating:** 2
**Confidence:** 5

**Summary:**

The paper replaces Brownian motion in diffusion models with fractional Brownian motion.

**Strengths:**

The mathematical description of fractional Brownian motion is clear.

**Weaknesses:**

1. The paper has very thin numerical experiments. The results do not adequately justify why replacing BM with fractional BM is preferred.

2. In order to show that fractional BM is better than BM, the experiments should focus on exactly showing the marginal improvement while controlling everything else, which the paper does not do.

3. The paper does not have strong methodology contribution, in additional to the replacement of fractional BM.

**Questions:**

If the authors really want to make some contributions to image generation, I suggest they spend more energy understanding the structure of image data, instead of focusing only on plugging in a fractional BM to replace BM.

**Limitations:**

yes

---

> ### Author Response · Authors · 2024-08-07
>
> We appreciate the reviewer's comments and feedback. However, we believe the strong criticism is excessively harsh and unjustified. The absence of a summary or acknowledgment of our method's strengths raises concerns about the reviewer's full understanding of our work. Below, we address the specific comments and questions they posed:
>
> >  thin numerical experiments.
>
> We now provide a set of additional experiments and report the computational considerations in the attached rebuttal PDF. We have included all of these in a further revision. We have also included a motivation for exploiting fractional Brownian motion (fBM) instead of Brownian motion (BM):
>
> Having control over the diffusion trajectories are crucial for obtaining a trade-off between generation quality and diversity. Moreover, there is no true support for why the noise increments should be independent. Our framework provides two hyperparameters, the number of processes and the Hurst index to have control over diffusion processes and characterize the nature of trajectories, \emph{e.g.} roughness vs. mild randomness. Our results indicate that in both aspects, quality and diversity, the diffusion models driven by Brownian motion lead to suboptimal results and the best outcome are obtained when $K\geq 2$ and $H>0.5$. This indicates that more regular and correlated trajectories are preferred while a single process is usually not sufficient to capture the entire complexity.
>
> > ... the experiments should focus on exactly showing the marginal improvement while controlling everything else,
>
> In fact, we exactly do this. We have two parameters, the number of Markov processes ($K$) and the Hurst index ($H$), that differentiate our method from a standard diffusion model. When $K=1$ and $H=0.5$, our method closely recovers the original model driven by Brownian motion. Our evaluations cover a range of $K$ and $H$ values, clearly demonstrating the improvements from our contributions. We have to admit that we suspect the reviewer might have overlooked some of the important parts in our paper.
>
> > The paper does not have strong methodology contribution, in additional to the replacement of fractional BM
>
> We kindly yet strongly disagree that *our method does not have a strong methodology contribution*. Replacing BM with fBM in diffusion models is definitely a non-trivial task due to the non-Markovian nature of fBM. To achieve this, we have made significant contributions, including deriving a memory and computation-efficient reverse process, proving the optimality of the score model, and providing explicit formulas for the marginals of the conditional forward process. Besides, strength of a contribution is a subjective matter and the reviewer does not mention on which aspects our contribution is not strong.
>
> > focus more on understanding the structure of image data rather than the noise
>
> Regrettably, this is another subjective comment. We agree that there are various ways to improve diffusion models. In this work, we specifically focuses on the driving noise, which, as shown by our experiments, has a significant impact in quality and diversity. Our work should be evaluated within this context. We find it surprising that the reviewer recommends a different, vague research direction instead of providing constructive feedback on our work.
>
> In the light of these and the feedback of other reviewers, we kindly ask the reviewer to reconsider their initial assessment.

---

> > ### Comment · Reviewer_kgmF · 2024-08-09
> >
> > It is interesting that the authors claim the review to be subjective, while they themselves argue in a subjective way with strong assumptions and vague arguments.
> >
> > As an example, the review pointed out that "the experiments should focus on exactly showing the marginal improvement while controlling everything else, which the paper does not do." The authors claim that they "exactly do so", by saying that "our method closely recovers the original model". This not exacting showing the marginal improvement. The controlling should be done way more precisely, say, on training-related parameters and hyper-parameters, all the randomness involved, etc. Just having "method closely recovers the original model" does not mean much, if anything. The authors just throw in a more complicated model and method and claim they "exactly do so" and controlled everything.
> >
> > This is an example of the authors being inconsistent. The authors do not understand what it means by "exactly showing the marginal improvement", while at the same time accusing the reviewer's understanding of the paper and "overlooked some of the important parts in our paper."
> >
> > Regarding other points in the authors' response:
> >
> > Methodological contribution: the point that the authors raised about replacing BM with fBM would be a non-trivial (subjective opinion of the authors, which others would find rather trivial) task does not merit strong methodological contribution. The work is simply replacing the BM with fBM, where the authors take "non-Markovian" to be a big point. Unfortunately, there is nothing fancy about non-Markovian, except for how long and how much it can depend on the past.
> >
> > The authors on one hand uses "strength of a contribution is a subjective matter" to defend themselves and on the other hand accuse the reviewer of not acknowledging the work's strengths. It is fairly convenient use of self-contracting standards when they see fit for themselves.
> >
> > The review suggested "focus more on understanding the structure of image data rather than the noise", which the authors accuse of being subjective. This review comment suggests the authors give more focus on how the fBM noise provides more understanding of the structure of image data when they are learned from denoising fBM noise. This is about providing more understanding of why fBM is essential, instead of focusing too much on the technical details of how to handle fBM. The authors took a very defensive and accusing attitude toward the review, with their own subjective understanding.

---

> ### Author Response · Authors · 2024-08-11
>
> As authors, we find it difficult to comprehend the perspective of this reviewer, and we believe that further engagement in what appears to be a non-scientific discussion would not be productive. We have brought this matter to the attention of the Area Chair and respectfully request that the other reviewers form their judgments independently of this review, which we believe to be biased.

---

> > ### Comment · Reviewer_kgmF · 2024-08-14
> >
> > The reviewer's comments include two parts: (1) pointing out the paper's need to exactly showing their marginal contribution by controlling other factors, without which the paper is purely a plug-in paper without verifiable contributions and (2) pointing out the authors' inconsistency with themselves and double standards regarding what is subjective.
> >
> > The authors responded to none of the reviewer's comments, and the authors again left a subjective message. It is unfortunate that the authors chose to run away from discussions when their own inconsistency and double standards are revealed.

---

### Official Review · Reviewer_jifC · 2024-07-15

**Soundness:** 3
**Presentation:** 3
**Contribution:** 3
**Rating:** 7
**Confidence:** 4

**Summary:**

The paper proposed diffusion models where Brownian motions are generalized by fractional Brownian motions - a correlated noise process. Due to the non-Markovian property of fractional Brownian motions, the paper suggests approximating it using a linear combination of semimartingale processes. The key insight of inducing fractional noise to diffusion models is to allow a control over the diversity sampling.  Experiments consider different setups of Hurst exponent parameters, demonstrating good performance in terms of inception score and Vendi score.

**Strengths:**

- The paper is well written, providing necessary background and come up the solution of variance exploding and variation preserving.
- The idea of fractional diffusion models is explored in-depth compared to Tong et al. 2022.

**Weaknesses:**

- I find the discussion in the experiments in the main text does not give much insight. What is the main message that the paper is trying to convey? What is the implication of the numbers?
- Missing the computational trade-off between Fractional Diffusion Models and traditional Diffusion models. The approximation using $K$ different semimartingale processes requires to sample these processes. How does it affect the training and inference?

**Questions:**

1. Can the Markovian approximation of fractional Brownian motion be extended to multifractional Brownian motions where the Hurst index varies with respect to time?
2. Can you discuss further potential applications of Fractional Diffusion Models? While the paper suggests they are promising for generating time series, especially financial time series, do you think this model could be applied to molecular design, given that atomic interactions follow fractional Brownian motions?

Minor points:

Missing table reference in Line 722.

**Limitations:**

Please see above.

---

> ### Author Rebuttal · Authors · 2024-08-07
>
> We thank the reviewer for their valuable feedback and truly insightful and thoughtful questions, which highlight the potential contributions of our work. We address specific comments and questions below:
>
> >I find the discussion in the experiments in the main text does not give much insight. What is the main message that the paper is trying to convey? What is the implication of the numbers?
>
> We agree with the reviewer that we did not clearly highlight the advantages of our framework and presented too many numbers that should have been included in the appendix alongside Table 5 and Table 6. We replace Table 1 and Table 2 in our paper with the attached Table 1, where we retrained our model on CIFAR10 on all configurations that we already investigated on MNIST. Comparing the two tables, we see a better performance of the super-diffusive regime ($H>0.5$) across the two datasets, reaching in both experiments the best FID for $H=0.9$, with either $3$ or $2$ augmenting processes. In Figure 3, it is seen that the best performance in terms of FID of the purely Brownian dynamics is dominated by the super-diffusive regime of our method, saturating on a lower FID level with aleary $500$ NFE. In addition, the pixel-wise diversity of the best performing configuration on CIFAR is higher than the pixel-wise diversity of the two baseline dynamics.
>
> > Missing the computational trade-off between Fractional Diffusion Models and traditional Diffusion models. The approximation using different semimartingale processes requires to sample these processes. How does it affect the training and inference?
>
> We commented on the computation time of GFDM in lines 229-232, arguing that the computational load increases only minimally. We show that the same number of score model evaluations is sufficient, where that the score-model has the same dimensionality as the input data and do not depend on the dimensionality of the augmented system. Since the score-model evaluation takes up the most compute time, augmenting the system with additional process increases the computation time only very little. In addition to this commentary, we now provide a quantitative evaluation of the computational trade-off between GFDM and traditional Diffusion Models.
>
> Table 3 in the attached pdf tabulates the average time GFDM needs to perform one training step for different $H$ and $K$.
> There is a mild increase of average compute time when switching from the original model to the augmented system, while no significant change between different Hurst indices $H$ or $K\leq5$.
>
> Table 4 in the attached pdf shows the average time GFDM needs to perform one sampling step. For $K\leq4$ the average compute time in seconds differs at most by 0.02 seconds, while there is a significant increase switching from $K=4$ to $K=5$. This is presumably due to the inversion of the covariance matrix of the augmenting processes needed to simulate the reverse time model.
>
> >Can the Markovian approximation of fractional Brownian motion be extended to multifractional Brownian motions where the Hurst index varies with respect to time?
>
> It is certainly conceivable that such an extension is possible. The form of the Markovian representation of fractional Brownian motion (Theorem A.2) suggests that this could be achieved by making the densities $\nu_1$ and $\nu_2$ depend on time. While a formal derivation and proof is beyond the scope of the work, this is a very interesting extension to be explored in the future. We now discuss this in our future work.
>
> >While the paper suggests they are promising for generating time series, especially financial time series, do you think this model could be applied to molecular design, given that atomic interactions follow fractional Brownian motions?
>
> This is a really great suggestion that we have in mind for our future work. Combining our framework with a Brownian Bridge model to switch between different unknown distributions might be a promising direction. In particular the reviewer's suggestion to use our framework for molecular evolution might be very promising, since noise increments being independent posits a strong assumption.
>
> We hope that we have sufficiently addressed the weaknesses and questions raised by the reviewer.
>
>
>
> With best regards,
>
> The Authors

---

> > ### Comment · Reviewer_jifC · 2024-08-13
> >
> > Thank you for the clarification. I believe that the paper is technically solid. Therefore, I maintain the original score.

---

### Author Rebuttal · Authors · 2024-08-07

Dear Reviewers and Respected Area Chair,

We would like to express our gratitude for your feedback.

We appreciate the reviewers' acknowledgement of the novelty of our method for the controlled use of fractional Brownian motion in diffusion models. We are also grateful for the appreciation reviewers shared for the clarity of our mathematical framework and the techniques underlying our method, such as the continuous reparameterization trick or augmented score matching to efficiently learn the score-function, that lead to tractable learning and inference, a D-dimensional score model, and marginal statistics for faster training, among other things.

Reviewers across the board requested a clearer concentration of the empirical takeaways. To address this concern, we restructured our section on experiments to give a complete overview of the ablations and configurations that we ran to benchmark the performance of our framework.

We replace Table 1 and Table 2 in our paper with Table 1 from the attached pdf, where we retrained our model on CIFAR10 on all configurations that we already investigated on MNIST. Comparing the two tables, we see a similar behaviour of GFDM accross the two datasets:

In the super-diffusive regime ($H>0.5$) we observe for $H=0.9$ on MNIST and and CIFAR10 the best performance in terms of FID providing on CIFAR10  as well a higher pixel-wise diversity compared to the purely Brownian driven dynamics.

In the attached Figure 1 we observe a similar pattern across different datasets and dynamics, indicating that more than one augmenting process yields the best performance.

Evaluating the performance with different number of sampling steps in Figure 3a) shows that the super-diffusive regime of MA-fBM saturates at 500 NFE on a lower level than the purely Brownian driven dynamics.

Additionally we give a quantitative evaluation of the computational trade-off comparing the average compute per training and sampling step of traditional diffusion models an GFDM ind Figure b) and Figure d).

Finally, answers to detailed questions can be found in the responses to each of your reviews.

We thank you again for your suggestions. Please let us know if you have any further questions.

---

### Decision · Program_Chairs · 2024-09-25

**Decision:**

Accept (poster)

**Comment:**

The paper proposes an extension of diffusion models where the Brownian motion driving the noise process is replaced by fractional Brownian motion. However, this introduces several challenges due to the non-Markovian nature of fractional Brownian motion. To address this problem, the paper suggests approximating this process using a linear combination of Ornstein–Uhlenbeck processes driven by the same Brownian motion. Experiments on standard image dataset benchmarks illustrate the method's behavior for different Hurst exponent parameters and demonstrate good performance.

Most of the reviewers and myself initially found the paper borderline. However, the authors addressed most of the concerns raised by the reviewers during the rebuttal. I trust they will incorporate the reviewers' feedback and the clarifications provided during the rebuttal into the final version of the paper.